# Minimax Optimization with Smooth Algorithmic Adversaries

**Tanner Fiez**,[*] **Lillian J. Ratliff**
University of Washington, Seattle
{fiezt, ratliffl}@uw.edu

**Chi Jin**
Princeton University
chij@princeton.edu

**Praneeth Netrapalli**
Google Research, India
pnetrapalli@google.com

## Abstract

This paper considers minimax optimization $\min_x \max_y f(x, y)$ in the challenging setting where $f$ can be both nonconvex in $x$ and nonconcave in $y$. Though such optimization problems arise in many machine learning paradigms including training generative adversarial networks (GANs) and adversarially robust models, from a theoretical point of view, two fundamental issues remain: (i) the absence of simple and efficiently computable optimality notions, and (ii) cyclic or diverging behavior of existing algorithms. This paper proposes a new theoretical framework for nonconvex-nonconcave minimax optimization that addresses both of the above issues. The starting point of this paper is the observation that, under a computational budget, the max-player can not fully maximize $f(x, \cdot)$ since nonconcave maximization is NP-hard in general. So, we propose a new framework, and a corresponding algorithm, for the min-player to play against *smooth algorithms* deployed by the adversary (i.e., the max-player) instead of against full maximization. Our algorithm is guaranteed to make monotonic progress (thus having no limit cycles or diverging behavior), and to find an appropriate "stationary point" in a polynomial number of iterations. Our framework covers practically relevant settings where the smooth algorithms deployed by the adversary are multi-step stochastic gradient ascent, and its accelerated version. We further present experimental results that confirm our theoretical findings and demonstrate the effectiveness of the proposed approach in practice on simple, conceptual settings.

## 1 Introduction

This paper considers minimax optimization $\min_x \max_y f(x, y)$ in the context of two-player zero-sum games, where the min-player (controlling $x$) tries to minimize objective $f$ assuming a worst-case opponent (controlling $y$) that acts so as to maximize it. Minimax optimization naturally arises in a variety of important machine learning paradigms, with the most prominent examples being the training of generative adversarial networks (GANs) (Goodfellow et al., 2014) and adversarially robust models (Madry et al., 2018). These applications commonly engage deep neural networks with various techniques such as convolution, recurrent layers, and batch normalization. As a result, the objective function $f$ is highly *nonconvex* in $x$ and *nonconcave* in $y$.

Theoretically, minimax optimization has been extensively studied starting from the seminal work of von Neumann (Neumann, 1928), with many efficient algorithms proposed for solving it (Robinson, 1951; Korpelevich, 1976; Nemirovski, 2004). A majority of these classical results have been focused on *convex-concave* functions, and heavily rely on the minimax theorem, i.e., $\min_x \max_y f(x, y) = \max_y \min_x f(x, y)$, which no longer holds beyond the convex-concave setting. Recent line of works (Lin et al., 2020a; Nouiehed et al., 2019; Thekumparampil et al., 2019; Lin et al., 2020b; Ostrovskii et al., 2020) address the *nonconvex-concave* setting where $f$ is nonconvex in $x$ but concave

---

[*]Currently with Amazon.

in $y$ by proposing meaningful optimality notions and designing algorithms to find such points. A key property heavily exploited in this setting is that the inner maximization over $y$ given a fixed $x$ can be computed efficiently, which unfortunately does not extend to the *nonconvex-nonconcave* setting.

Consequently, nonconvex-nonconcave optimization remains challenging, and two of the most fundamental theoretical issues are still unresolved: (i) what is an appropriate notion of optimality that can be computed efficiently? and (ii) can we design algorithms that do not suffer from cycling or diverging behavior? Practitioners often use simple and popular algorithms such as gradient descent ascent (GDA) and other variants for solving these challenging optimization problems. While these algorithms seem to perform well in terms of producing high quality images in GANs and robust models in adversarial training, they are also highly unstable, particularly in training GANs. Indeed the instability of GDA and other empirically popular methods is not surprising since they are known to not converge even in very simple settings (Daskalakis & Panageas, 2018a; Bailey et al., 2020). This current state of affairs strongly motivates the need to develop a strong theoretical foundation for nonconvex-nonconcave minimax optimization and to design better algorithms for solving them.

**This work** considers the challenging nonconvex-nonconcave setting. Our framework sprouts from the practical consideration that under a computational budget, the max-player cannot fully maximize $f(x, \cdot)$ since nonconcave maximization is NP-hard in general. In fact, in both the settings of GAN and adversarial training, in practice, the max-player employs simple gradient based algorithms such as gradient ascent, run for a few steps – on the order of $5$ for GAN training and $40$ for adversarial training (see, e.g., Arjovsky et al. 2017; Madry et al. 2018) – to estimate $\mathrm{argmax}_y\, f(x, y)$. To capture this aspect, we assume that the max-player has a toolkit of multiple (potentially randomized) algorithms $\mathcal{A}_1, \mathcal{A}_2, \cdots, \mathcal{A}_k$ in an attempt to solve the maximization problem given fixed $x$, and picks the best solution among these algorithms. This motivates us to study the surrogate of the minimax optimization problem as

$$\min_x \max_{i \in [k]} f(x, \mathcal{A}_i(x)) = \min_x \max_{\lambda \in \Delta_k} \sum_{i=1}^k \lambda_i f(x, \mathcal{A}_i(x)), \tag{1}$$

where $\Delta_k$ denotes the $k$-dimensional simplex, and $\mathcal{A}_i(x)$ denotes the output of algorithm $\mathcal{A}_i$ for a given $x$. When both the objective function $f$ and the algorithms $\{\mathcal{A}_i\}_{i=1}^k$ are smooth (defined formally in Section 3), we can show that (1) becomes a *smooth nonconvex-concave* minimax optimization problem, where recent advances can be leveraged in solving such problems.

In particular, given the smooth algorithms deployed by the adversary (i.e. the max-player), this paper proposes two algorithms for solving problems in (1). The first algorithm is based on stochastic gradient descent (SGD), which is guaranteed to find an appropriate notion of "$\epsilon$-approximate stationary point" in $\mathcal{O}(\epsilon^{-4})$ gradient computations. The second algorithm is based on proximal algorithm, in the case of *deterministic* adversarial algorithms $\{\mathcal{A}_i\}_{i=1}^k$, this algorithm has an improved gradient complexity $\mathcal{O}(\epsilon^{-3})$ or $\tilde{\mathcal{O}}(\mathrm{poly}(k)/\epsilon^2)$ depending on the choice of subroutine within the algorithm. All our algorithms are guaranteed to make monotonic progress, thus having no limit cycles.

Our second set of results show that, many popular algorithms deployed by the adversary such as multi-step stochastic gradient ascent, and multi-step stochastic Nesterov's accelerated gradient ascent are in fact smooth. Therefore, our framework readily applies to those settings in practice.

We also present complementing experimental results showing that our theoretical framework and algorithm succeed on simple, conceptual examples of GAN and adversarial training. While more work is needed to scale our approach to large scale benchmarks in GAN and adversarial training, our experimental results serve as a proof of concept demonstration that our algorithm converges to desirable points in practice, at least on simple conceptual examples.

## 2   RELATED WORK

Due to lack of space, we focus our discussion here on directly related works and present a more detailed overview of related work in Appendix A. Many existing works that propose local optimality notions in nonconvex-nonconcave minimax optimization suffer from nonexistence of such points in general (Ratliff et al., 2013; Ratliff et al., 2016; Jin et al., 2020; Fiez et al., 2020; Zhang et al., 2020a; Farnia & Ozdaglar, 2020). To the best of our knowledge, the only works that propose a relaxed local optimality notion that is shown to exist and be computable in polynomial time is due to Keswani et al. (2020); Mangoubi & Vishnoi (2021). The aforementioned works are similar to this

paper in the sense that the min-player faces the max-player with computational restrictions, but are different from ours in terms of the model of the max-player and the algorithms to solve the problem. Concretely, Keswani et al. (2020); Mangoubi & Vishnoi (2021) consider the stationary points of a "smoothed" greedy-max function, which is computed by maximizing along a locally ascending path when fixing the min-player. In contrast, our paper considers the stationary points of a "max" function computed by particular smooth algorithms given their initializations, which is a more faithful surrogate to the objective that the min-player wishes to minimize. Depending on the choice of smooth algorithms and their initializations, the optimal points of Keswani et al. (2020); Mangoubi & Vishnoi (2021) and the optimal points in our paper can be very different, making them incomparable.

## 3 PRELIMINARIES

In this section, we present problem formulation and preliminaries. We consider function $f$ satisfying

**Assumption 1.** *We denote $w = (x, y)$, and assume $f : \mathbb{R}^{d_1} \times \mathbb{R}^{d_2} \to \mathbb{R}$ is:*

    *(a) B-bounded i.e., $|f(w)| \leq B$,      (b) G-Lipschitz i.e., $|f(w_1) - f(w_2)| \leq G\|w_1 - w_2\|$,*

    *(c) L-gradient Lipschitz i.e., $\|\nabla f(w_1) - \nabla f(w_2)\| \leq L\|w_1 - w_2\|$,*

    *(d) $\rho$-Hessian Lipschitz i.e., $\|\nabla^2 f(w_1) - \nabla^2 f(w_2)\| \leq \rho\|w_1 - w_2\|$.*

*where $\|\cdot\|$ denotes Euclidean norm for vectors and operator norm for matrices.*

We aim to solve $\min_{x \in \mathbb{R}^{d_1}} \max_{y \in \mathbb{R}^{d_2}} f(x, y)$. Since $\max_{y \in \mathbb{R}^{d_2}} f(x, y)$ involves non-concave maximization and hence is NP-hard in the worst case, we intend to play against algorithm(s) that $y$-player uses to compute her strategy. Concretely, given $x \in \mathbb{R}^{d_1}$, we assume that the $y$-player chooses her (potentially random) strategy $\widehat{y}_z(x) = \mathcal{A}_{i^*(x)}(x, z_{i^*(x)})$, where we use shorthand $z := (z_1, \cdots, z_k)$, as $i^*(x) = \operatorname{argmax}_{i \in [k]} f(x, \mathcal{A}_i(x, z_i))$, where $\mathcal{A}_1, \cdots, \mathcal{A}_k$ are $k$ *deterministic* algorithms that take as input $x$ and a random seed $z_i \in \mathbb{R}^\ell$, where $z_i$ are all independent. Note that the framework captures randomized algorithms e.g., $\mathcal{A}$ could be stochastic gradient ascent on $f(x, \cdot)$, with initialization, mini-batching etc. determined by the random seed $z$. This also incorporates running the same algorithm multiple times, with different seeds and then choosing the best strategy. We now reformulate the minimax objective function to:

$$\min_{x \in \mathbb{R}^{d_1}} g(x) \quad \text{where} \quad g(x) := \mathbb{E}_z \left[ f(x, \widehat{y}_z(x)) \right]. \tag{2}$$

For general algorithms $\mathcal{A}_i$, the functions $f(x, \mathcal{A}_i(x, z_i))$ need not be continuous even when $f$ satisfies Assumption 1. However, if the algorithms $\mathcal{A}_i$ are smooth as defined below, the functions $f(x, \mathcal{A}_i(x, z_i))$ behave much more nicely.

**Definition 1** (Algorithm Smoothness). *A randomized algorithm $\mathcal{A} : \mathbb{R}^{d_1} \times \mathbb{R}^\ell \to \mathbb{R}^{d_2}$ is:*

    *(a) G-Lipschitz, if $\|\mathcal{A}(x_1, z) - \mathcal{A}(x_2, z)\| \leq G\|x_1 - x_2\|$ for any $z$.*

    *(b) L-gradient Lipschitz, if $\|D\mathcal{A}(x_1, z) - D\mathcal{A}(x_2, z)\| \leq L\|x_1 - x_2\|$ for any $z$.*

Here $D\mathcal{A}(x, z) \in \mathbb{R}^{d_1} \times \mathbb{R}^{d_2}$ is the Jacobian of the function $\mathcal{A}(\cdot, z)$ for a fixed $z$. The following lemma tells us that $f(x, \mathcal{A}(x, z))$ behaves nicely whenever $\mathcal{A}$ is a Lipschitz and gradient Lipschitz algorithm. For deterministic algorithms, we also use the shortened notation $\mathcal{A}(x)$ and $D\mathcal{A}(x)$.

**Lemma 1.** *Suppose $\mathcal{A}$ is $G'$-Lipschitz and $L'$-gradient Lipschitz and $f$ satisfies Assumption 1. Then, for a fixed $z$, function $f(\cdot, \mathcal{A}(\cdot, z))$ is $G(1 + G')$-Lipschitz and $L(1 + G')^2 + GL'$-gradient Lipschitz.*

While $g(x)$ defined in (2) is not necessarily gradient Lipschitz, it can be shown to be weakly-convex as defined below. Note that an $L$-gradient Lipschitz function is $L$-weakly convex.

**Definition 2.** *A function $g : \mathbb{R}^{d_1} \to \mathbb{R}$ is $L$-weakly convex if $\forall\, x$, there exists a vector $u_x$ satisfying:*

$$g(x') \geq g(x) + \langle u_x, x' - x \rangle - \frac{L}{2}\|x' - x\|^2 \quad \forall\, x'. \tag{3}$$

*Any vector $u_x$ satisfying this property is called the subgradient of $g$ at $x$ and is denoted by $\nabla g(x)$.*

An important property of weakly convex function is that the maximum over a finite number of weakly convex function is still a weakly convex function.

**Lemma 2.** *Given $L$-weakly convex functions $g_1, \cdots, g_k : \mathbb{R}^d \to \mathbb{R}$, the maximum function $g(\cdot) := \max_{i \in [k]} g_i(\cdot)$ is also $L$-weakly convex and the set of subgradients of $g(\cdot)$ at $x$ is given by:*

$$\partial g(x) = \{\textstyle\sum_{j \in S(x)} \lambda_j \nabla g_j(x) : \lambda_j \geq 0,\ \sum_{j \in S(x)} \lambda_j = 1\}, \text{where } S(x) := \operatorname{argmax}_{i \in [k]} g_i(x).$$

Thus, under Assumption 1 and the assumptions $\mathcal{A}_i$ are all $G'$-Lipschitz and $L'$-gradient Lipschitz, then $g(\cdot)$ defined in (2) is $L(1 + G')^2 + GL'$-weakly convex. The usual optimality notion for weakly-convex functions is approximate first order stationary points (Davis & Drusvyatskiy, 2018).

**Approximate first-order stationary point for weakly convex functions**: In order to define approximate stationary points, we also need the notion of Moreau envelope.

**Definition 3.** *The* Moreau envelope *of a function* $g : \mathbb{R}^{d_1} \to \mathbb{R}$ *and parameter* $\lambda$ *is:*

$$g_\lambda(x) \;=\; \min_{x' \in \mathbb{R}^{d_1}} g(x') + (2\lambda)^{-1} \|x - x'\|^2 . \tag{4}$$

The following lemma provides useful properties of the Moreau envelope.

**Lemma 3.** *For an* $L$-*weakly convex function* $g : \mathbb{R}^{d_1} \to \mathbb{R}$ *and* $\lambda < 1/L$, *we have:*

  (a) *The minimizer* $\hat{x}_\lambda(x) = \arg\min_{x' \in \mathbb{R}^{d_1}} g(x') + (2\lambda)^{-1} \|x - x'\|^2$ *is unique and* $g(\hat{x}_\lambda(x)) \leq g_\lambda(x) \leq g(x)$. *Furthermore,* $\arg\min_x g(x) = \arg\min_x g_\lambda(x)$.

  (b) $g_\lambda$ *is* $\lambda^{-1}(1 + (1 - \lambda L)^{-1})$-*smooth and thus differentiable, and*

  (c) $\min_{u \in \partial g(\hat{x}_\lambda(x))} \|u\| \leq \lambda^{-1} \|\hat{x}_\lambda(x) - x\| = \|\nabla g_\lambda(x)\|$.

First order stationary points (FOSP) of a non-smooth nonconvex function are well-defined, i.e., $x^*$ is a *FOSP* of a function $g(x)$ if, $0 \in \partial f(x^*)$. However, unlike smooth functions, it is nontrivial to define an *approximate* FOSP. For example, if we define an $\varepsilon$-FOSP as the point $x$ with $\min_{u \in \partial g(x)} \|u\| \leq \varepsilon$, where $\partial g(x)$ denotes the subgradients of $g$ at $x$, there may never exist such a point for sufficiently small $\varepsilon$, unless $x$ is exactly a FOSP. In contrast, by using above properties of the Moreau envelope of a weakly convex function, it's approximate FOSP can be defined as (Davis & Drusvyatskiy, 2018):

**Definition 4.** *Given an* $L$-*weakly convex function* $g$, *we say that* $x^*$ *is an* $\varepsilon$-*first order stationary point* ($\varepsilon$-FOSP) *if,* $\|\nabla g_{1/2L}(x^*)\| \leq \varepsilon$, *where* $g_{1/2L}$ *is the Moreau envelope with parameter* $1/2L$.

Using Lemma 3, we can show that for any $\varepsilon$-FOSP $x^*$, there exists $\hat{x}$ such that $\|\hat{x} - x^*\| \leq \varepsilon/2L$ and $\min_{u \in \partial g(\hat{x})} \|u\| \leq \varepsilon$. In other words, an $\varepsilon$-FOSP is $O(\varepsilon)$ close to a point $\hat{x}$ which has a subgradient smaller than $\varepsilon$. Other notions of FOSP proposed recently such as in Nouiehed et al. (2019) can be shown to be a strict generalization of the above definition.

## 4 MAIN RESULTS

In this section, we present our main results. Assuming that the adversary employs Lipschitz and gradient-Lipschitz algorithms (Assumption 2), Section 4.1 shows how to compute (stochastic) subgradients of $g(\cdot)$ (defined in (2)) efficiently. Section 4.2 further shows that stochastic subgradient descent (SGD) on $g(\cdot)$ can find an $\epsilon$-FOSP in $O\left(\epsilon^{-4}\right)$ iterations while for the deterministic setting, where the adversary uses only deterministic algorithms, Section 4.3 provides a proximal algorithm that can find an $\epsilon$-FOSP faster than SGD. For convenience, we denote $g_{z,i}(x) := f(x, \mathcal{A}_i(x, z_i))$ and recall $g(x) := \mathbb{E}_z \left[ \max_{i \in [k]} g_{z,i}(x) \right]$. For deterministic $\mathcal{A}_i$, we drop $z$ and just use $g_i(x)$.

### 4.1 COMPUTING STOCHASTIC SUBGRADIENTS OF $g(x)$

In this section, we give a characterization of subgradients of $g(x)$ and show how to compute stochastic subgradients efficiently under the following assumption.

**Assumption 2.** *Algorithms* $\mathcal{A}_i$ *in* (2) *are* $G'$-*Lipschitz and* $L'$-*gradient Lipschitz as per Definition 1.*

Under Assumptions 1 and 2, Lemma 1 tells us that $g_{z,i}(x)$ is a $G(1+G')$-Lipschitz and $L(1+G')^2 + GL'$-gradient Lipschitz function for every $i \in [k]$ with

$$\nabla g_{z,i}(x) = \nabla_x f(x, \mathcal{A}_i(x, z_i)) + D\mathcal{A}_i(x, z_i) \cdot \nabla_y f(x, \mathcal{A}_i(x, z_i)), \tag{5}$$

where we recall that $D\mathcal{A}_i(x, z_i) \in \mathbb{R}^{d_1 \times d_2}$ is the Jacobian matrix of $\mathcal{A}_i(\cdot, z_i) : \mathbb{R}^{d_1} \to \mathbb{R}^{d_2}$ at $x$ and $\nabla_x f(x, \mathcal{A}_i(x, z_i))$ denotes the partial derivative of $f$ with respect to the first variable at $(x, \mathcal{A}_i(x, z_i))$. While there is no known general recipe for computing $D\mathcal{A}_i(x, z_i)$ for an arbitrary algorithm $\mathcal{A}_i$, most algorithms used in practice such as stochastic gradient ascent (SGA), stochastic Nesterov accelerated gradient (SNAG), ADAM, admit efficient ways of computing these derivatives e.g., *higher* package

---

**Algorithm 1:** Stochastic subgradient descent (SGD)

---

**Input:** initial point $x_0$, step size $\eta$

1 **for** $s = 0, 1, \ldots, S$ **do**
2      Sample $z_1, \cdots, z_k$ and compute $\widehat{\nabla} g(x_s)$ according to eq. (6).
3      $x_{s+1} \leftarrow x_s - \eta \widehat{\nabla} g(x_s)$.
4 **return** $\bar{x} \leftarrow x_s$, where $s$ is uniformly sampled from $\{0, \cdots, S\}$.

---

in PyTorch (Grefenstette et al., 2019). For concreteness, we obtain expression for gradients of SGA and SNAG in Section 5 but the principle behind the derivation holds much more broadly and can be extended to most algorithms used in practice (Grefenstette et al., 2019). In practice, the cost of computing $\nabla g_{z,i}(x)$ in (5) is at most twice the cost of evaluating $g_{z,i}(x)$—it consists a forward pass for evaluating $g_{z,i}(x)$ and a backward pass for evaluating its gradient (Grefenstette et al., 2019).

Lemma 2 shows that $g(x) := \mathbb{E}_z \left[ \max_{i \in [k]} g_{z,i}(x) \right]$ is a weakly convex function and a stochastic subgradient of $g(\cdot)$ can be computed by generating a random sample of $z_1, \cdots, z_k$ as:

$$\widehat{\nabla} g(x) = \sum_{j \in S(x)} \lambda_j \nabla g_{z,i}(x) \text{ for any } \lambda \in \Delta_k, \text{ where } S(x) := \underset{i \in [k]}{\operatorname{argmax}} \, g_{z,i}(x) \tag{6}$$

Here $\Delta_k$ is the $k$-dimensional probability simplex. It can be seen that $\mathbb{E}_z[\hat{\nabla} g(x)] \in \partial g(x)$. Furthermore, if all $\mathcal{A}_i$ are deterministic algorithms, then the above is indeed a subgradient of $g$.

### 4.2 CONVERGENCE RATE OF SGD

The SGD algorithm to solve (2) is given in Algorithm 1. The following theorem shows that Algorithm 1 finds an $\epsilon$-FOSP of $g(\cdot)$ in $S = O\left(\epsilon^{-4}\right)$ iterations. Since each iteration of Algorithm 1 requires computing the gradient of $g_{z,i}(x)$ for each $i \in [k]$ at a single point $x_s$, this leads to a total of $S = O\left(\epsilon^{-4}\right)$ gradient computations for each $g_{z,i}(x)$.

**Theorem 1.** *Under Assumptions 1 and 2, if $S \geq 16B\widehat{L}\widehat{G}^2\epsilon^{-4}$ and learning rate $\eta = (2B/[\widehat{L}\widehat{G}^2(S+1)])^{1/2}$ then output of Algorithm 1 satisfies $\mathbb{E}[\|\nabla g_{1/2\widehat{L}}(\bar{x})\|^2] \leq \epsilon^2$, where $\widehat{L} := L(1+G')^2 + GL'$ and $\widehat{G} := G(1+G')$.*

Theorem 1 claims that the expected norm squared of the Moreau envelope gradient of the output point satisfies the condition for being an $\epsilon$-FOSP. This, together with Markov's inequality, implies that at least half of $x_0, \cdots, x_S$ are $2\epsilon$-FOSPs, so that the probability of outputting a $2\epsilon$-FOSP is at least 0.5. Proposition 1 in Appendix C shows how to use an efficient postprocessing mechanism to output a $2\epsilon$-FOSP with high probability. Theorem 1 essentially follows from the results of (Davis & Drusvyatskiy, 2018), where the key insight is that, in expectation, the SGD procedure in Algorithm 1 almost monotonically decreases the Moreau envelope evaluated at $x_s$ i.e., $\mathbb{E}\left[g_{1/2\widehat{L}}(x_s)\right]$ is almost monotonically decreasing. This shows that Algorithm 1 makes (almost) monotonic progress in a precise sense and hence does not have limit cycles. In contrast, none of the other existing algorithms for nonconvex-nonconcave minimax optimization enjoy such a guarantee. Note that this claim includes extra-gradient and optimistic gradient methods that have been touted as empirically mitigating cycling behavior; see Appendix F for examples demonstrating nonconvergence of these methods.

### 4.3 PROXIMAL ALGORITHM WITH FASTER CONVERGENCE FOR DETERMINISTIC ALGORITHMS

While the rate achieved by SGD is the best known for weakly-convex optimization with a stochastic subgradient oracle, faster algorithms exist for functions which can be written as *maximum over a finite number of smooth functions* with *access to exact subgradients of these component functions*. These conditions are satisfied when $\mathcal{A}_i$ are all deterministic and satisfy Assumption 2. A pseudocode of such a fast proximal algorithm, inspired by Thekumparampil et al. (2019, Algorithm 3), is presented in Algorithm 2. However, in contrast to the results of Thekumparampil et al. (2019), the following theorem provides two alternate ways of implementing Step 3 of Algorithm 2, resulting in two different (and incomparable) convergence rates.

---

**Algorithm 2:** Proximal algorithm

---

**Input:** initial point $x_0$, target accuracy $\epsilon$, smoothness parameter $\widehat{L}$

**1** $\widehat{\epsilon} \leftarrow \epsilon^2/(64\widehat{L})$

**2 for** $t = 0, 1, \ldots, S$ **do**

**3** | Find $x_{s+1}$ such that

$$\max_{i \in [k]} g_i(x_{s+1}) + \widehat{L}\|x_s - x_{s+1}\|^2 \leq \left(\min_x \max_{i \in [k]} g_i(x) + \widehat{L}\|x_s - x\|^2\right) + \widehat{\epsilon}/4$$

**4** | **if** $\max_{i \in [k]} g_i(x_{s+1}) + \widehat{L}\|x_s - x_{s+1}\|^2 \geq \max_{i \in [k]} g_i(x_s) - 3\widehat{\epsilon}/4$ **then**

**5** | | **return** $x_s$

---

**Theorem 2.** *Under Assumptions 1 and 2, if $\widehat{L} := L(1 + G')^2 + GL' + kG(1 + G')$ and $S \geq 200\widehat{L}B\epsilon^{-2}$ then Algorithm 2 returns $x_s$ satisfying $\|\nabla g_{1/2\widehat{L}}(x_s)\| \leq \epsilon$. Depending on whether we use (Thekumparampil et al., 2019, Algorithm 1) or cutting plane method (Lee et al., 2015) for solving Step 3 of Algorithm 2, the total number of gradient computations of each $g_i$ is $\widetilde{O}\left(\frac{\widehat{L}^2 B}{\epsilon^3}\right)$ or $O\left(\frac{\widehat{L}B}{\epsilon^2} \cdot \mathrm{poly}(k) \log \frac{\widehat{L}}{\epsilon}\right)$ respectively.*

Ignoring the parameters $L, G, L'$ and $G'$, the above theorem tells us that Algorithm 2 outputs an $\epsilon$-FOSP using $O\left(k\epsilon^{-3}\right)$ or $O\left(\mathrm{poly}(k)\epsilon^{-2} \log \frac{1}{\epsilon}\right)$ gradient queries to each $g_i$ depending on whether Thekumparampil et al. (2019, Algorithm 3) or cutting plane method (Lee et al., 2015) was used for implementing Step 3. While the proximal algorithm itself works even when $\mathcal{A}_i$ are randomized algorithms, there are no known algorithms that can implement Step (3) with fewer than $O\left(\epsilon^{-2}\right)$ stochastic gradient queries. Hence, this does not improve upon the $O\left(\epsilon^{-4}\right)$ guarantee for Algorithm 1 when $\mathcal{A}_i$ are randomized algorithms. The proof of Theorem 2 shows that the iterates $x_s$ monotonically decrease the value $g(x_s)$, guaranteeing that there are no limit cycles for Algorithm 2 as well.

## 5 SMOOTHNESS OF POPULAR ALGORITHMS

In this section, we show that two popular algorithms—namely, $T$-step stochastic gradient ascent (SGA) and $T$-step stochastic Nesterov's accelerated gradient ascent (SNAG)—are both Lipschitz and gradient-Lipschitz satisfying Assumption 2 and hence are captured by our results in Section 4. Consider the setting $f(x, y) = \frac{1}{n} \sum_{j \in [n]} f_j(x, y)$. Let $z$ be a random seed that captures the randomness in the initial point as well as minibatch order in SGA and SNAG. We first provide the smoothness results on $T$-step SGA for different assumptions on the shape of the function $f$ and for $T$-step SNAG. After giving these results, we make remarks interpreting their significance and the implications.

$T$**-step SGA:** For a given $x$ and random seed $z$, the $T$-step SGA update is given by: $y_{t+1} = y_t + \eta \nabla_y f_{\sigma(t)}(x, y_t)$, where $\sigma : [T] \to [N]$ is a sample selection function and $\eta$ is the stepsize. Observe that with the same randomness $z$, the initial point does not depend on $x$ i.e., $y_0(x) = y_0(x')$, so $Dy_0 = 0$. The following theorems provide the Lipschitz and gradient Lipschitz constants of $y_T(x)$ (as generated by $T$-step SGA) for the general nonconvex-nonconcave setting as well as the settings in which the function $f$ is nonconvex-concave and nonconvex-strongly concave.

**Theorem 3** (General Case). *Suppose for all $j \in [n]$, $f_j$ satisfies Assumption 1. Then, for any fixed randomness $z$, $T$-step SGA is $(1 + \eta L)^T$-Lipschitz and $4(\rho/L) \cdot (1 + \eta L)^{2T}$-gradient Lipschitz.*

**Theorem 4** (Concave Case). *Suppose for all $j \in [n]$, $f_j$ satisfies Assumption 1 and $f_j(x, \cdot)$ is concave for any $x$. Then, for any fixed randomness $z$, $T$-step SGA is $\eta LT$-Lipschitz and $(\rho/L) \cdot (1 + \eta LT)^3$-gradient Lipschitz.*

**Theorem 5** (Strongly-concave Case). *Suppose for all $j \in [n]$, $f_j$ satisfies Assumption 1 and $f_j(x, \cdot)$ is $\alpha$-strongly concave for any $x$. Then, for any fixed randomness $z$, $T$-step SGA is $\kappa$-Lipschitz and $4(\rho/L) \cdot \kappa^3$-gradient Lipschitz, where $\kappa = L/\alpha$ is the condition number.*

$T$**-step SNAG:** For a given random seed $z$, the $T$-step SNAG update is given by:

$$\tilde{y}_t = y_t + (1 - \theta)(y_t - y_{t-1}) \quad \text{and} \quad y_{t+1} = \tilde{y}_t + \eta \nabla_y \widehat{f}_{\sigma(t)}(x, \tilde{y}_t),$$

where $\eta$ is the stepsize, $\theta \in [0, 1]$ is the momentum parameter. The output of the algorithm is given by $\mathcal{A}(x, z) = y_T(x)$. Furthermore, we have the following guarantee.

**Theorem 6** (General Case). *Suppose for all $j \in [n]$, $f_j$ satisfies Assumption 1. Then, for any fixed seed $z$, $T$-step SNAG is $T(1 + \eta L/\theta)^T$-Lipschitz and $50(\rho/L) \cdot T^3(1 + \eta L/\theta)^{2T}$-gradient Lipschitz.*

**Remarks on the Impact of the Smoothness Results:** The Lipschitz and gradient Lipschitz parameters for $T$-step SGA and $T$-step SNAG in the setting where $f$ is nonconvex-*nonconcave* are all exponential in $T$, the duration of the algorithm. In general, this seems unavoidable in the worst case for the above algorithms and also seems to be the case for most of the other popular algorithms such as ADAM, RMSProp, etc. In contrast, in the nonconvex-*concave* and nonconvex-*strongly concave* settings, our results show that the smoothness parameters of $T$-step SGA are no longer exponential in $T$. In particular, in the nonconvex-*concave* the Lipschitz parameter is linear in $T$ and the gradient Lipschitz parameter is polynomial in $T$ while in the nonconvex-*strongly concave*, the analogous smoothness parameters are no longer dependent on $T$. We conjecture this is also the case for $T$-step SNAG, though the proof appears quite tedious and hence, we opted to leave that for future work. For problems of practical importance, however, we believe that the smoothness parameters are rarely exponential in $T$. Our experimental results confirm this intuition (see Figure 2). Furthermore, note that the function $g(x)$ (2), and its stationary points, vary as one varies $T$. While in adversarial training, $T$ is essentially determined by the power of adversary against which we wish the model to be robust, in GAN training, $T$ is actually an algorithmic knob that can be tuned to improve the final generator performance. Indeed, larger $T$ does not necessarily lead to better generators and small values of $T$ are of vital interest; as is pointed out in the original paper (Goodfellow et al., 2014), fully optimizing the discriminator can result in overfitting and too strong of a discriminator can limit the ability of the generator to learn. Finally, while we prove the Lipschitz and gradient Lipschitz properties only for SGA and SNAG, we believe that the same techniques could be used to prove similar results for popular algorithms such as ADAM, RMSProp, etc. However, there are other algorithms, particularly those involving projection that are not gradient Lipschitz (see Proposition 3 in Appendix E.5).

## 6  EMPIRICAL RESULTS

This section presents empirical results evaluating our SGD algorithm (Algorithm 1) for generative adversarial networks (Goodfellow et al., 2014) and adversarial training (Madry et al., 2018). Our results demonstrate that our framework results in stable monotonic improvement during training and converges to desirable solutions in both GAN and adversarial training problems. While optimization through the algorithm of the adversary increases the complexity of each update by a factor of two, our results demonstrate that it also leads to faster convergence. Finally, we show the gradient norms do not grow exponentially in the number of gradient ascent steps $T$ taken by the adversary in practice.

**Generative Adversarial Networks.** Generative adversarial network formulations can be characterized by the following minimax problem (Nagarajan & Kolter, 2017; Mescheder et al., 2018):

$$\min_\theta \max_\omega f(\theta, \omega) = \mathbb{E}_{x \sim p_\mathcal{X}}[\ell(D_\omega(x))] + \mathbb{E}_{z \sim p_\mathcal{Z}}[\ell(-D_\omega(G_\theta(z)))]. \tag{7}$$

In this formulation $G_\theta : \mathcal{Z} \rightarrow \mathcal{X}$ is the generator network parameterized by $\theta$ that maps from the latent space $\mathcal{Z}$ to the input space $\mathcal{X}$, $D_\omega : \mathcal{X} \rightarrow \mathbb{R}$ is discriminator network parameterized by $\omega$ that maps from the input space $\mathcal{X}$ to real-valued logits, and $p_\mathcal{X}$ and $p_\mathcal{Z}$ are the distributions over the input and latent spaces. The loss function defines the objective where $\ell(w) = -\log(1 + \exp(-w))$ gives the original "saturating" generative adversarial networks formulation (Goodfellow et al., 2014).

**Dirac–GAN.** The Dirac-GAN (Mescheder et al., 2018) is a simple and common baseline for evaluating the efficacy of generative adversarial network training methods. In this problem, the generator distribution $G_\theta(z) = \delta_\theta$ is a Dirac distribution concentrated at $\theta$, the discriminator network $D_\omega(x) = -\omega x$ is linear, and the real data distribution $p_\mathcal{X} = \delta_0$ is a Dirac distribution concentrated at zero. The resulting objective after evaluating (7) with the loss $\ell(w) = -\log(1 + \exp(-w))$ is

$$\min_\theta \max_\omega f(\theta, \omega) = \ell(\theta\omega) + \ell(0) = -\log(1 + e^{-\theta\omega}) - \log(2).$$

To mimic the real data distribution, the generator parameter $\theta$ should converge to $\theta^* = 0$. Notably, simultaneous and alternating gradient descent-ascent are known to cycle and fail to converge on this problem (see Figure 1). We consider an instantiation of our framework where the discriminator

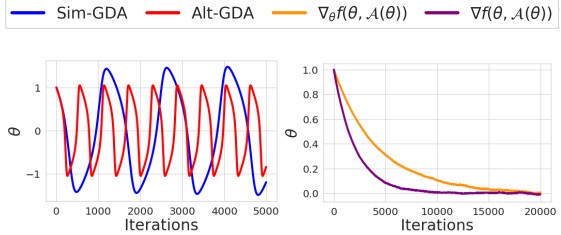
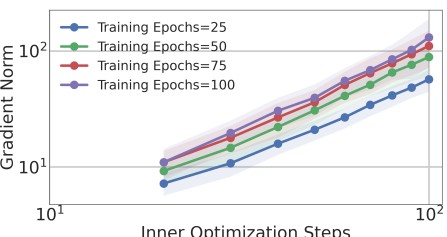

Figure 1: **Dirac-GAN**: Generator parameters while training using simultaneous and alternating gradient descent-ascent (left), and our framework (right) with & without optimizing through the discriminator. Under our framework, training is stable and converges to correct distribution. Further, differentiating through the discriminator results in faster convergence.

Figure 2: **Adversarial training**: $\|\nabla f(\theta, \mathcal{A}(\theta))\|$ as a function of number of steps $T$ taken by gradient ascent (GA) algorithm $\mathcal{A}$ evaluated at multiple points in the training procedure. The plot shows that, in practice, the Lipschitz parameter of GA does not grow exponentially in $T$.

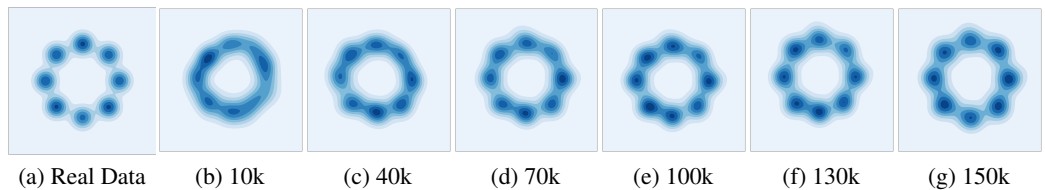

| (a) Real Data | (b) 10k | (c) 40k | (d) 70k | (e) 100k | (f) 130k | (g) 150k |

Figure 3: **Mixture of Gaussians**: Generated distribution at various steps during course of training. We see that training is stable and results in monotonic progress towards the true distribution.

samples an initialization uniformly between $[-0.1, 0.1]$ and performs $T = 10$ steps of gradient ascent between each generator update. The learning rates for both the generator and the discriminator are $\eta = 0.01$. We present the results in Figure 1. Notably, the generator parameter monotonically converges to the optimal $\theta^* = 0$ and matches the real data distribution using our training method. We also show the performance when the generator descends using partial gradient $\nabla_\theta f(x, \mathcal{A}(\theta))$ instead of the total gradient $\nabla f(\theta, \mathcal{A}(\theta))$ in Algorithm 1. This method is able to converge to the optimal generator distribution but at a slower rate. Together, this example highlights that our method fixes the usual cycling problem by reinitializing the discriminator and also that optimizing through the discriminator algorithm is key to fast convergence. Additional results are given in Appendix G.

**Mixture of Gaussians.** We now demonstrate that the insights we developed from the Dirac-GAN (stability and monotonic improvement) carry over to the more complex problem of learning a 2-dimensional mixture of Gaussians. This is a common example and a number of papers (see, e.g., Metz et al. 2017; Mescheder et al. 2017; Balduzzi et al. 2018) show that standard training methods using simultaneous or alternating gradient descent-ascent can fail. The setup for the problem is as follows. The real data distribution consists of 2-dimensional Gaussian distributions with means given by $\mu = [\sin(\phi), \cos(\phi)]$ for $\phi \in \{k\pi/4\}_{k=0}^{7}$ and each with covariance $\sigma^2 I$ where $\sigma^2 = 0.05$. For training, the real data $x \in \mathbb{R}^2$ is drawn at random from the set of Gaussian distributions and the latent data $z \in \mathbb{R}^{16}$ is drawn from a standard normal distribution with batch sizes of 512. The network for the generator and discriminator contain two and one hidden layers respectively, each of which contain 32 neurons and ReLU activation functions. We consider the objective from (7) with $\ell(w) = -\log(1 + \exp(-w))$ which corresponds to the "saturating" generative adversarial networks formulation (Goodfellow et al., 2014). This objective is known to be difficult to train since with typical training methods the generator gradients saturate early in training.

We show results using our framework in Figure 3 where the discriminator performs $T = 15$ steps of gradient ascent and the initialization between each generator step is obtained by the default network initialization in Pytorch. The generator and discriminator learning rates are both fixed to be $\eta = 0.5$. We see that our method has stable improvement during the course of training and recovers close to the real data distribution. We demonstrate in Appendix G that this result is robust by presenting the

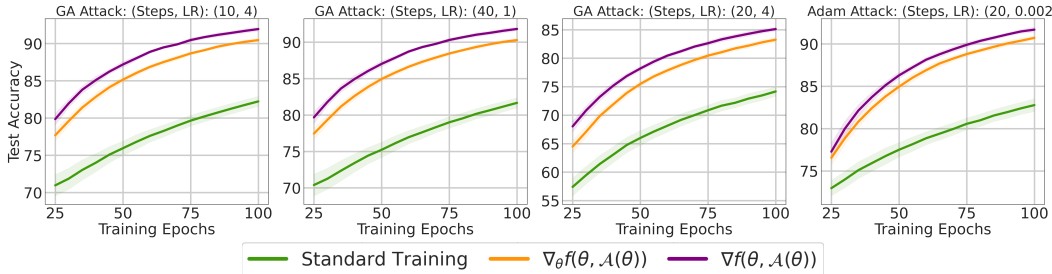

Figure 4: **Adversarial training**: Test accuracy during course of training where the attack used during training is gradient ascent (GA) with learning rate (LR) of $4$ and number of steps (Steps) of $10$ but evaluated against attacks with different Steps and LR. These plots show that training with a single attack gives more robustness to even other attacks with different parameters or algorithms of similar computational power. This empirical insight is complementary to our theoretical results that extend to the setting of training against multiple smooth algorithms. Finally, using total gradient $\nabla f(\theta, \mathcal{A}(\theta))$ yields better robustness compared to using partial gradient $\nabla_\theta f(\theta, \mathcal{A}(\theta))$.

final output of 10 runs of the procedure. Notably, the training algorithm recovers all the modes of the distribution in each run. We also show results using Adam for the discriminator in Appendix G.

**Adversarial Training.** Given a data distribution $\mathcal{D}$ over pairs of examples $x \in \mathbb{R}^d$ and labels $y \in [k]$, parameters $\theta$ of a neural network, a set $\mathcal{S} \subset \mathbb{R}^d$ of allowable adversarial perturbations, and a loss function $\ell(\cdot, \cdot, \cdot)$ dependent on the network parameters and the data, adversarial training amounts to considering a minmax optimization problem of the form $\min_\theta \mathbb{E}_{(x,y)\sim\mathcal{D}}[\max_{\delta\in\mathcal{S}} \ell(\theta, x + \delta, y)]$. In practice (Madry et al., 2018), the inner maximization problem $\max_{\delta\in\mathcal{S}} \ell(\theta, x + \delta, y)$ is solved using *projected* gradient ascent. However, as described in Section 5, this is not a smooth algorithm and does not fit our framework. So, we use gradient ascent, *without projection*, for the inner maximization.

We run an adversarial training experiment with the MNIST dataset, a convolutional neural network, and the cross entropy loss function. We compare Algorithm 1 with usual adversarial training (Madry et al., 2018) which descends $\nabla_\theta f(\theta, \mathcal{A}(\theta))$ instead of $\nabla f(\theta, \mathcal{A}(\theta))$, and a baseline of standard training without adversarial training. For each algorithm, we train for 100 passes over the training set using a batch size of 50. The minimization procedure has a fixed learning rate of $\eta_1 = 0.0001$ and the maximization procedure runs for $T = 10$ steps with a fixed learning rate of $\eta_2 = 4$. We evaluate the test classification accuracy during the course of training against gradient ascent or Adam optimization adversarial attacks. The results are presented in Figure 4 where the mean accuracies are reported over 5 runs and the shaded regions show one standard deviation around the means. We observe that the adversarial training procedure gives a significant boost in robustness compared to standard training. Moreover, consistent with the previous experiments, our algorithm which uses total gradient outperforms standard adversarial training which uses only partial gradient. We present results against more attacks in Appendix G. As suggested in Section 5, we also find that in practice, the gradient norms $\|\nabla f(\theta, \mathcal{A}(\theta))\|$ do not grow exponentially in the number of gradient ascent steps $T$ in the adversary algorithm $\mathcal{A}$ (see Figure 2). For further details and additional results see Appendix G.

## 7 CONCLUSION

Motivated by the facts that (i) nonconcave maximization is NP-hard in the worst case and (ii) in practice, simple gradient based algorithms are used for maximization, this paper presents a new framework for solving nonconvex-nonconcave minimax optimization problems. Assuming that the min player has knowledge of the *smooth* algorithms being used by max player, we propose new efficient algorithms and verify that these algorithms find desirable solutions in practice on small-scale generative adversarial network and adversarial training problems. Furthermore, the framework proposed in this paper extends naturally to two player non-zero sum games as well. An important future direction of work is to scale our algorithm to large scale, state of the art GAN benchmarks. The key challenge here is that reinitializing the discriminator parameters necessitates running several steps of gradient ascent for discriminator for every generator update. One approach to overcome this could be to use pretrained networks for initializing the discriminator.

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

## A   DETAILED RELATED WORK

**Nonconvex-Nonconcave Zero-Sum Games.** The existing work on nonconvex-nonconcave zero-sum games has generally focused on (1) defining and characterizing local equilibrium solution concepts and (2) analyzing the local stability and convergence behavior of gradient-based learning algorithms around fixed points of the dynamics. The concentration on local analysis stems from the inherent challenges that arise in nonconvex-nonconcave zero-sum games from both a dynamical systems perspective and a computational perspective. In particular, it is know that broad classes of gradient-based learning dynamics can admit limit cycles and other non-trivial periodic orbits that are antithetical to any type of global convergence guarantee in this class of games (Hsieh et al., 2020; Letcher, 2021). Moreover, on constrained domains, it has been shown that finding even a local equilibrium is computationally intractable (Daskalakis et al., 2021).

A number of local equilibrium notions for nonconvex-nonconcave zero-sum games now exist with characterizations in terms of gradient-based conditions relevant to gradient-based learning. This includes the local Nash (Ratliff et al., 2013; Ratliff et al., 2016) and local minmax (Stackelberg) (Jin et al., 2020; Fiez et al., 2020) equilibrium concepts, which both amount to local refinements and characterizations of historically standard game-theoretic equilibrium notions. In terms of provable guarantees, algorithms incorporating higher-order gradient information have been proposed and analyzed that guarantee local convergence to only local Nash equilibria (Adolphs et al., 2019; Mazumdar et al., 2019) or local convergence to only local minmax equilibria (Wang et al., 2020; Zhang et al., 2020b; Fiez et al., 2020) in nonconvex-nonconcave zero-sum games. Beyond the local Nash and minmax equilibrium, notions including the proximal equilibrium concept (Farnia & Ozdaglar, 2020), which is a class between the set of local Nash and local minmax equilibria, and the local robust equilibrium concept (Zhang et al., 2020a), which includes both local minmax and local maxmin equilibria, have been proposed and studied. It is worth noting that a shortcoming of each of the local equilibrium notions is that may fail to exist on unconstrained domains.

Significant attention has been given to the local stability and convergence of simultaneous gradient descent-ascent in nonconvex-nonconcave zero-sum games. This stems from the fact that it is the natural analogue of learning dynamics for zero-sum game optimization to gradient descent for function optimization. Moreover, simultaneous gradient descent-ascent is know to often perform reasonably well empirically and is ubiquitous in a number of applications such as in training generative adversarial networks and adversarial learning. However, it has been shown that while local Nash are guaranteed to be stable equilibria of simultaneous gradient descent-ascent (Mazumdar et al., 2020; Daskalakis & Panageas, 2018b; Jin et al., 2020), local minmax may not be unless there is sufficient timescale separation between the minimizing and maximizing players (Jin et al., 2020; Fiez & Ratliff, 2021). Specific to generative adversarial networks, it has been shown that simultaneous gradient descent-ascent locally converges to local equilibria under certain assumptions on the generator network and the data distribution (Nagarajan & Kolter, 2017; Mescheder et al., 2018). Later in this section we discuss in further detail learning dynamics studied previously in games which bear resemblance to that which we consider in this paper depending on the model of the maximizing player.

The challenges of nonconvex-nonconcave zero-sum games we have highlighted limit the types of provable guarantees that can be obtained and consequently motivate tractable relaxations including to nonconvex-concave zero-sum games and the general framework we formulate in this work. Before moving on, we mention that from a related perspective, a line of recent work (Keswani et al., 2020; Mangoubi & Vishnoi, 2021) in nonconvex-nonconcave zero-sum games proposes relaxed equilibrium notions that are shown to be computable in polynomial time and are guaranteed to exist. At a high level, the equilibria correspond to a joint strategy at which the maximizing player is at an approximate local maximum of the cost function and the minimizing player is at an approximate local minimum of a smoothed and relaxed best-response function of the maximizing player. The aforementioned works are similar to this paper in the sense that the minimizing player faces a maximizing player with computational restrictions, but diverge in terms of the model of the maximizing player and the algorithms for solving the problem.

**Nonconvex-Concave Zero-Sum Games.** The past few years has witnessed a significant amount of work on gradient-based dynamics in nonconvex-concave zero-sum games. The focus of existing work on nonconvex-concave zero-sum games has key distinctions from that in nonconvex-nonconcave

zero-sum games. Generally, the work on nonconvex-concave zero-sum games has analyzed dynamics on constrained domains, where typically the strategy space of the maximizing player is constrained to a closed convex set and occasionally the minimizing player also faces a constraint. In contrast, nonconvex-nonconcave zero-sum games have generally been analyzed on unconstrained domains. Moreover, instead of focusing on computing notions of game-theoretic equilibrium as is typical in nonconvex-nonconcave zero-sum games, the body of work on nonconvex-concave zero-sum games has focused on achieving stationarity of the game cost function $f(\cdot, \cdot)$ or the best-response function $\Phi(\cdot) = \max_y f(\cdot, y)$.

The structure present in nonconvex-concave zero-sum games has been shown to simplify the problem compared to nonconvex-nonconcave zero-sum games so that global finite-time convergence guarantees are achievable. Thus, work in this direction has focused on improving the rates of convergence in terms of the gradient complexity to find $\epsilon$–approximate stationary points of $f(\cdot, \cdot)$ or $\Phi(\cdot)$, both with deterministic and stochastic gradients. Guarantees on the former notion of stationarity can be translated to guarantees on the latter notion of stationarity with extra computational cost (Lin et al., 2020a).

For the the class of nonconvex-strongly-concave zero-sum games, a series of works design algorithms that are shown to obtain $\epsilon$–approximate stationary points of the functions $f(\cdot, \cdot)$ or $\Phi(\cdot)$ with a gradient complexity of $\widetilde{O}(\epsilon^{-2})$ in terms of $\epsilon$ in the deterministic setting (Jin et al., 2020; Rafique et al., 2021; Lu et al., 2020; Lin et al., 2020a;b). In the deterministic nonconvex-strongly concave problem, the notions of stationarity are equivalent in terms of the dependence on $\epsilon$ up to a logarithmic dependence (Lin et al., 2020a). Lower bounds for this problem have also been established (Zhang et al., 2021; Li et al., 2021). In the stochastic nonconvex-strongly-concave problem, existing work has developed algorithms that are shown to obtain $\epsilon$–approximate stationary points of the function $\Phi(\cdot)$ in gradient complexities of $\widetilde{O}(\epsilon^{-4})$ (Rafique et al., 2021; Jin et al., 2020; Lin et al., 2020a) and $\widetilde{O}(\epsilon^{-3})$ (Luo et al., 2020) in terms of $\epsilon$ dependence.

In the deterministic nonconvex-concave problem, a number of algorithms with provable guarantees to $\epsilon$–approximate stationary points of the function $f(\cdot, \cdot)$ have been shown with gradient complexities of $\widetilde{O}(\epsilon^{-4})$ (Lu et al., 2020), $\widetilde{O}(\epsilon^{-3.5})$ (Nouiehed et al., 2019), and $\widetilde{O}(\epsilon^{-2.5})$ (Ostrovskii et al., 2020; Lin et al., 2020b). Similarly, in this class of problems, there exist results on algorithms that guarantee convergence to an $\epsilon$–approximate stationary points of the function $\Phi(\cdot)$ with gradient complexities $\widetilde{O}(\epsilon^{-6})$ (Rafique et al., 2021; Jin et al., 2020; Lin et al., 2020a) and $\widetilde{O}(\epsilon^{-3})$ (Thekumparampil et al., 2019; Zhao, 2020; Kong & Monteiro, 2019; Lin et al., 2020b). Finally, existing results in the stochastic setting for achieving an $\epsilon$–approximate stationary point of $\Phi(\cdot)$ show gradient complexities of $\widetilde{O}(\epsilon^{-6})$ (Rafique et al., 2021) and $\widetilde{O}(\epsilon^{-8})$ (Lin et al., 2020a).

In this work, we build on the developments for nonconvex-concave problems to obtain our results.

**Gradient-Based Learning with Opponent Modeling.** A number of gradient-based learning schemes have been derived is various classes of games based on the following idea: if a player knows how the opponents in a game are optimizing their cost functions, then it is natural to account for this behavior in the players own optimization procedure. The simultaneous gradient descent learning dynamics can be viewed as the simplest instantiation of this perspective, where each player is optimizing their own cost function assuming that all other players in the game will remain fixed. In general, the more sophisticated existing learning dynamics based on opponent modeling assume the opponents are doing gradient descent on their cost function and this prediction is incorporated into the objective being optimized in place of the current strategies of opponents. A key conceptual distinction between this approach and our work is that in existing opponent modeling methods the dynamics of the players are always updated simultaneously whereas the procedure we consider is sequential in nature with the opponent initializing again at each interaction. Moreover, the types of guarantees we prove are distinct compared to existing work in this realm.

In this modern literature, gradient-based learning with opponent modeling dates back to the work of Zhang & Lesser (2010). They study simple two-player, two-action, general-sum matrix games, and analyze a set of learning dynamics called iterated descent descent with policy prediction (IGA-PP) and show asymptotic convergence to a Nash equilibrium. In this set of learning dynamics, each player assumes the other player is doing gradient descent and this prediction is used in the objective. In particular, each player $i$ has a choice variable $x^i$ and a cost function $f_i(x^i, x^{-i})$ that after

incorporating the prediction becomes $f_i(x_t^i, x_t^{-i} - \gamma \nabla_{-i} f_{-i}(x_t^i, x_t^{-i}))$. To optimize the objective, each player takes a first-order Taylor expansion of their cost function to give the augmented objective

$$f_i(x_t^i, x_t^{-i} - \gamma \nabla_{-i} f_{-i}(x_t^i, x_t^{-i})) \approx f_i(x_t^i, x_t^{-i}) - \gamma \nabla_{-i} f_i(x_i^t, x_{-i}^t)^\top \nabla_{-i} f_{-i}(x_t^i, x_t^{-i}).$$

Each player in the game simultaneously follows the gradient of their augmented objective which is given by

$$\nabla_i f_i(x_t^i, x_t^{-i}) - \gamma \nabla_{-i,i} f_i(x_i^t, x_{-i}^t)^\top \nabla_{-i} f_{-i}(x_t^i, x_t^{-i}).$$

This gradient computation is derived based on the fact that the assumed update of the other player $\nabla_{-i} f_{-i}(x_t^i, x_t^{-i})$ does not depend on the optimization variable.

Similar ideas have recently been revisited in more general nonconvex multiplayer games (Foerster et al., 2018; Letcher et al., 2019). In learning with opponent learning awareness (LOLA) (Foerster et al., 2018), players again assume the other players are doing gradient descent and take their objective to be $f_i(x_t^i, x_t^{-i} - \gamma \nabla_{-i} f_{-i}(x_t^i, x_t^{-i}))$. To derive the learning rule, an augmented objective is again formed by computing a first-order Taylor expansion, but now the term $\nabla_{-i} f_{-i}(x_t^i, x_t^{-i})$ in the augmented objective is treated as dependent on the optimization variable so that the gradient of the augmented objective is given by

$$\nabla_i f_i(x_t^i, x_t^{-i}) - \gamma \nabla_{-i,i} f_i(x_i^t, x_{-i}^t)^\top \nabla_{-i} f_{-i}(x_t^i, x_t^{-i}) - \gamma \nabla_{-i,i} f_{-i}(x_t^i, x_t^{-i})^\top \nabla_{-i} f_i(x_i^t, x_{-i}^t).$$

Finally, to arrive at the final gradient update for each player, the middle term in the equation above is removed and each player takes steps along the gradient update

$$\nabla_i f_i(x_t^i, x_t^{-i}) - \gamma \nabla_{-i,i} f_{-i}(x_t^i, x_t^{-i})^\top \nabla_{-i} f_i(x_i^t, x_{-i}^t).$$

While no convergence results are given for LOLA, a follow-up work shows local convergence guarantees to stable fixed points for IGA-PP and learning dynamics called stable opponent shaping (SOS) that interpolate between IGA-PP and LOLA (Letcher et al., 2019). A related work derives learning dynamic based on the idea that the opponent selects a best-response to the chosen strategy (Fiez et al., 2020). The resulting learning dynamics can be viewed as LOLA with the opponent selecting a Newton learning rate. For nonconvex-nonconcave zero-sum games, local convergence guarantees to only local Stackelberg equilibrium are given in for this set of learning dynamics (Fiez et al., 2020). It is worth remarking that gradient-based learning with opponent modeling is historically rooted in the general framework of consistent conjectural variations (see, e.g., Başar & Olsder 1998, Chapter 4.6), a concept that is now being explored again and is closely related to the previously mentioned learning dynamics (Chasnov et al., 2020).

Perhaps the closest work on gradient-based learning with opponent modeling to this paper is that of unrolled generative adversarial networks (Metz et al., 2017). In unrolled generative adversarial networks, the generator simulates the discriminator doing a fixed number of gradient steps from the current parameter configurations of the generator and discriminator. The resulting discriminator parameters are then used in place of the current discriminator parameters in the generator objective. The generator then updates following the gradient of this objective, optimizing through the rolled out discriminator update by computing the total derivative. Simultaneously with the generator update, the discriminator updates its parameters by performing a gradient step on its objective. In our framework, for generative adversarial networks when the discriminator is modeled as performing $T$-steps of gradient ascent, the procedure we propose is similar but an important difference is that when the generator simulates the discriminator unrolling procedure the discriminator parameters are initialized from scratch and there is no explicit discriminator being trained simultaneously with the generator.

**Extragradient methods**: Starting from the seminal work of Korpelevich (1976), extragradient methods have been studied extensively for solving minimax optimization problems. While such methods have been shown to obtain faster convergence rates (Nemirovski, 2004) and final iterate convergence (Daskalakis et al., 2018; Lei et al., 2021) for *convex-concave* minimax optimization, there is no result showing that it converges in general nonconvex-nonconcave settings. In fact, in Appendix F we present simple examples demonstrating that the extragradient method as well as another variant called optimistic gradient descent ascent, *do not converge* for general nonconvex-nonconcave minimax optimization.

**Games with computationally bounded adversaries**: There are also a few works in the game theory literature which consider resource/computationally bounded agents. For example Freund et al. (1995)

considers *repeated* games between resource bounded agents, (Sandhlom & Lesser, 1997) considers coalition formation between resource bounded agents in cooperative games and Halpern et al. (2014) shows that resource constraints in otherwise rational players might lead to some commonly observed human behaviors while making decisions. However, the settings, models of limited computation and the focus of results considered in all of these prior works are distinct from those of this paper.

**Stability of algorithms in numerical analysis**: To our knowledge such results on the smoothness of the classes of algorithms we study—i.e., gradient-based updates such as SGA and SNAG—with respect to problem parameters (e.g., in this case, $x$) have not been shown in the machine learning and optimization literature. This being said, in the study of dynamical systems—more specifically differential equations—the concept of continuity (and Lipschitzness) with respect to parameters and initial data has been studied using a *variational* approach wherein the continuity of the solution of the differential equation is shown to be continuous with respect to variations in the parameters or initial data by appealing to nonlinear variation of parameters results such as the Bellman-Grownwall inequality or Alekseev's theorem (see classical references on differential equations such as Coddington & Levinson (1955, Chapter 2) or Hartman (2002, Chapter IV.2)).

In numerical methods, such results on the "smoothness" or continuity of the differential equation with respect to initial data or problem parameters are used to understand stability of particular numerical methods (see, e.g., Atkinson et al. 2011, Chapter 1.2). In particular, a initial value problem is only considered well-posed if there is continuous dependence on initial data. For instance, the simple scalar differential equation

$$\dot{y}(t) = -y(t) + 1, \ 0 \le t \le T, \ y(0) = 1$$

has solution $y(t) \equiv 1$, yet the perturbed problem,

$$\dot{y}_\epsilon(t) = -y_\epsilon(t) + 1, \ 0 \le t \le T, \ y_\epsilon(0) = 1 + \epsilon,$$

has solution $y_\epsilon(t) = 1 + \epsilon e^{-t}$ so that

$$|y(t) - y_\epsilon(t)| \le |\epsilon|, \ 0 \le t \le T.$$

If the maximum error $\|y_\epsilon - y\|_\infty$ is (much) larger than $\epsilon$ then the initial value problem is *ill-conditioned* and any typical attempt to numerically solve such a problem will lead to large errors in the computed solution. In short, the stability properties of a numerical method (i.e., discretization of the differential equation) are fundamentally connected to the continuity (smoothness) with respect to initial data.

Observe that methods such as gradient ascent can be viewed as a discretization of an differential equation:

$$\dot{y}(t) = \nabla_y f(x, y(t)) \ \longrightarrow \ y_{k+1} = y_k + \eta \nabla_y f(x, y_k).$$

As such, the techniques for showing continuity of the solution of a differential equation with respect to initial data or other problem parameters (e.g., in this case $x$) can be adopted to show smoothness of the $T$-step solution of the discretized update. Our approach to showing smoothness, on the other hand, leverages the recursive nature of the discrete time updates defining the classes of algorithms we study. This approach simplifies the analysis by directly going after the smoothness parameters using the update versus solving the difference (or differential) equation for $y_T(x)$ and then finding the smoothness parameters which is the method typically used in numerical analysis of differential equations. An interesting direction of future research is to more formally connect the stability analysis from numerical analysis of differential equations to robustness of adversarial learning to initial data and even variations in problem parameters.

## B  PROOF OF RESULTS IN SECTION 3

*Proof of Lemma 1.* For any fixed $z$, we note that $\mathcal{A}(\cdot, z)$ is a deterministic algorithm. Consequently, it suffices to prove the lemma for a deterministic algorithm $\mathcal{A}(\cdot)$. By chain rule, the derivative of $f(x, \mathcal{A}(x))$ is given by:

$$\nabla f(x, \mathcal{A}(x)) = \nabla_x f(x, \mathcal{A}(x)) + D\mathcal{A}(x) \cdot \nabla_y f(x, \mathcal{A}(x)), \tag{8}$$

where $D\mathcal{A}(x) \in \mathbb{R}^{d_1 \times d_2}$ is the derivative of $\mathcal{A}(\cdot) : \mathbb{R}^{d_1} \to \mathbb{R}^{d_2}$ at $x$ and $\nabla_x f(x, \mathcal{A}(x))$ and $\nabla_y f(x, \mathcal{A}(x))$ denote the partial derivatives of $f$ with respect to the first and second variables

respectively at $(x, \mathcal{A}(x))$. An easy computation shows that

$$\|\nabla f(x, \mathcal{A}(x))\| \leq \|\nabla_x f(x, \mathcal{A}(x))\| + \|D\mathcal{A}(x)\| \cdot \|\nabla_y f(x, \mathcal{A}(x))\|$$
$$\leq G + G' \cdot G = (1 + G')G.$$

This shows that $f(x, \mathcal{A}(x))$ is $(1 + G')G$-Lipschitz. Similarly, we have:

$$\|\nabla f(x_1, \mathcal{A}(x_1)) - \nabla f(x_2, \mathcal{A}(x_2))\|$$
$$\leq \|\nabla_x f(x_1, \mathcal{A}(x_1)) - \nabla_x f(x_2, \mathcal{A}(x_2))\| + \|D\mathcal{A}(x_1)\nabla_y f(x_1, \mathcal{A}(x_1)) - D\mathcal{A}(x_2)\nabla_y f(x_2, \mathcal{A}(x_2))\|.$$

For the first term, we have:

$$\|\nabla_x f(x_1, \mathcal{A}(x_1)) - \nabla_x f(x_2, \mathcal{A}(x_2))\|$$
$$\leq \|\nabla_x f(x_1, \mathcal{A}(x_1)) - \nabla_x f(x_2, \mathcal{A}(x_1))\| + \|\nabla_x f(x_2, \mathcal{A}(x_1)) - \nabla_x f(x_2, \mathcal{A}(x_2))\|$$
$$\leq L(\|x_1 - x_2\| + \|\mathcal{A}(x_1) - \mathcal{A}(x_2)\|) \leq L(1 + G')\|x_1 - x_2\|.$$

Similarly, for the second term we have:

$$\|D\mathcal{A}(x_1)\nabla_y f(x_1, \mathcal{A}(x_1)) - D\mathcal{A}(x_2)\nabla_y f(x_2, \mathcal{A}(x_2))\|$$
$$\leq \|D\mathcal{A}(x_1)\|\|\nabla_y f(x_1, \mathcal{A}(x_1)) - \nabla_y f(x_2, \mathcal{A}(x_2))\| + \|\nabla_y f(x_2, \mathcal{A}(x_2))\|\|D\mathcal{A}(x_2) - D\mathcal{A}(x_1)\|$$
$$\leq (LG'(1 + G') + GL')\|x_1 - x_2\|.$$

This proves the lemma. $\qquad\square$

*Proof of Lemma 2.* Given any $x$ and $y$, and any $\lambda$ such that $\lambda_j \geq 0$ and $\sum_{j \in S(x)} \lambda_j = 1$, we have:

$$g(y) = \max_{j \in [k]} g_j(y) \geq \sum_{j \in S(x)} \lambda_j g_j(y) \geq \sum_{j \in S(x)} \lambda_j \left( g_j(x) + \langle \nabla g_j(x), y - x \rangle - \frac{1}{2L}\|x - y\|^2 \right)$$
$$= g(x) + \langle \sum_{j \in S(x)} \lambda_j \nabla g_j(x), y - x \rangle - \frac{1}{2L}\|x - y\|^2,$$

proving the lemma. $\qquad\square$

*Proof of Lemma 3.* We re-write $f_\lambda(x)$ as minimum value of a $(\frac{1}{\lambda} - L)$-strong convex function $\phi_{\lambda,x}$, as $g$ is $L$-weakly convex (Definition 2) and $\frac{1}{2\lambda}\|x - x'\|^2$ is differentiable and $\frac{1}{\lambda}$-strongly convex,

$$g_\lambda(x) = \min_{x' \in \mathbb{R}^{d_1}} \left[ \phi_{\lambda,x}(x') = g(x') + \frac{1}{2\lambda}\|x - x'\|^2 \right]. \tag{9}$$

Then first part of (a) follows trivially by the strong convexity. For the second part notice the following,

$$\min_x g_\lambda(x) = \min_x \min_{x'} g(x') + \frac{1}{2\lambda}\|x - x'\|^2$$
$$= \min_{x'} \min_x g(x') + \frac{1}{2\lambda}\|x - x'\|^2$$
$$= \min_{x'} g(x')$$

Furthermore, we will show that $\arg\min_x g_\lambda(x) = \arg\min_x g(x)$. Consider $x^* \in \operatorname{argmin}_x g(x)$. We have that for any $x$,

$$g_\lambda(x) = \min_{x'} g(x') + \frac{1}{2\lambda}\|x - x'\|^2 \geq g(x^*) \geq g_\lambda(x^*).$$

This shows that $x^* \in \operatorname{argmin}_x g_\lambda(x)$ and hence $\operatorname{argmin}_x g(x) \subseteq \operatorname{argmin}_x g_\lambda(x)$. Similarly, for any $x^* \in \operatorname{argmin}_x g_\lambda(x)$, we have:

$$g_\lambda(x^*) = g(\hat{x}_\lambda(x^*)) + \frac{1}{2\lambda}\|\hat{x}_\lambda(x^*) - x^*\|^2 \geq g(\hat{x}_\lambda(x^*)) \geq g_\lambda(\hat{x}_\lambda(x^*)).$$

---

**Algorithm 3:** Estimating Moreau envelope's gradient for postprocessing

---

**Input:** point $x$, stochastic subgradient oracle for function $g$, error $\epsilon$, failure probability $\delta$

1  Find $\widetilde{\mathbf{x}}$ such that $g(\widetilde{\mathbf{x}}) + L\|x - \widetilde{\mathbf{x}}\|^2 \leq \left(\min_{x'} g(x') + L\|x - x'\|^2\right) + \frac{\epsilon^2}{4L}$ using (Harvey et al., 2019, Algorithm 1).

2  **return** $2L(x - \widetilde{\mathbf{x}})$.

---

Since $x^* \in \operatorname{argmin}_x g_\lambda(x)$, the above inequality is infact an equality and hence $\hat{x}_\lambda(x^*) = x^*$. So, $g_\lambda(x^*) = g(x^*)$ and hence $x^* \in \operatorname{argmin}_x g(x)$ implying that $\operatorname{argmin}_x g_\lambda(x) \subseteq \operatorname{argmin}_x g(x)$. Thus $\arg\min_x g_\lambda(x) = \arg\min_x g(x)$. For $(b)$ we can re-write the Moreau envelope $g_\lambda$ as,

$$
\begin{aligned}
g_\lambda(x) &= \min_{x'} g(x') + \frac{1}{2\lambda}\|x - x'\|^2 \\
&= \frac{\|x\|^2}{2\lambda} - \frac{1}{\lambda}\max_{x'}(x^T x' - \lambda g(x') - \frac{\|x'\|^2}{2}) \\
&= \frac{\|x\|^2}{2\lambda} - \frac{1}{\lambda}\left(\lambda g(\cdot) + \frac{\|\cdot\|^2}{2}\right)^*(x)
\end{aligned}
\tag{10}
$$

where $(\cdot)^*$ is the Fenchel conjugation operator. Since $L < 1/\lambda$, using $L$-weak convexity of $g$, it is easy to see that $\lambda g(x') + \frac{\|x'\|^2}{2}$ is $(1 - \lambda L)$-strongly convex, therefore its Fenchel conjugate would be $\frac{1}{(1-\lambda L)}$-smooth (Kakade et al., 2009, Theorem 6). This, along with $\frac{1}{\lambda}$-smoothness of first quadratic term implies that $g_\lambda(x)$ is $\left(\frac{1}{\lambda} + \frac{1}{\lambda(1-\lambda L)}\right)$-smooth, and thus differentiable.

For $(c)$ we proceed as follows. Since $\hat{x}_\lambda(x) = \operatorname{argmin}_{x'}\phi_{\lambda,x}(x')$, by first order KKT optimality conditions, we have that

$$
x - \hat{x}_\lambda(x) \in \lambda\partial g(x).
\tag{11}
$$

Further, from proof of part (a) we have that $\phi_{\lambda,x}(x')$ $(1 - \lambda L)$-strongly-convex in $x'$ and it is quadratic (and thus convex) in $x$. Danskin's theorem (Bertsekas, 2009, Section 6.11) thus implies that for $g_\lambda(x) = \min_{x' \in \mathbb{R}^{d_1}} \phi_{\lambda,x}(x')$, we have that $\nabla g_\lambda(x) = (x - \hat{x}_\lambda(x))/\lambda$. Letting $\hat{u}_\lambda(x) := \lambda^{-1}(x - \hat{x}_\lambda(x)) \in \partial g(x)$, we have that:

$$
\min_{u \in \partial g(\hat{x}_\lambda(x))} \|u\| \leq \|\hat{u}_\lambda(x)\| = \lambda^{-1}\|x - \hat{x}_\lambda(x)\| = \|\nabla g_\lambda(x)\|.
$$

$\square$

## C    PROOFS OF RESULTS IN SECTION 4.2

In order to prove convergence of this algorithm, we first recall the following result from (Davis & Drusvyatskiy, 2018).

**Theorem 7** (Corollary 2.2 from (Davis & Drusvyatskiy, 2018))**.** *Suppose $g(\cdot)$ is $L$-weakly convex, and $\mathbb{E}_{z_1,\cdots,z_k}\left[\|\widehat{\nabla} g(x)\|^2\right] \leq G^2$. Then, the output $\bar{x}$ of Algorithm 1 with stepsize $\eta = \frac{\gamma}{\sqrt{S+1}}$ satisfies:*

$$
\mathbb{E}\left[\|\nabla g_{\frac{1}{2L}}(\bar{x})\|^2\right] \leq 2 \cdot \frac{\left(g_{\frac{1}{2L}}(x_0) - \min_x g(x)\right) + LG^2\gamma^2}{\gamma\sqrt{S+1}}.
$$

*Proof of Theorem 1.* Lemmas 1 and 2 tell us that $g(x)$ is $\widehat{L}$-weakly convex and for any choice of $z$, the stochastic subgradient $\widehat{\nabla} g(x)$ is bounded in norm by $\widehat{G}$. Consequently, Theorem 7 with the stated choice of $S$ proves Theorem 1. $\square$

**Proposition 1.** *Given a point $x$, an $L$-weakly convex function $g$ and a $G$-norm bounded and a stochastic subgradient oracle to $g$ (i.e., given any point $x'$, which returns a stochastic vector $u$ such that $\mathbb{E}[u] \in \partial g(x')$ and $\|u\| \leq G$), with probability at least $1 - \delta$, Algorithm 3 returns a vector $u$ satisfying $\|u - \nabla g_\lambda(x)\| \leq \epsilon$ with at most $O\left(\frac{G^2 \log\frac{1}{\delta}}{\epsilon^2}\right)$ queries to the stochastic subgradient oracle of $g$.*

*Proof of Proposition 1.* Let $\Phi_{\frac{1}{2L}}(x', x) := g(x') + L\|x - x'\|^2$. Recall the notation of Lemma 3 $\widehat{x}_{\frac{1}{2L}}(x) := \arg\min_{x'} \Phi_{\frac{1}{2L}}(x', x)$ and $g_\lambda(x) = \min_{x'} \Phi_{\frac{1}{2L}}(x', x)$. The proof of Lemma 3 tells us that $\nabla g_{\frac{1}{2L}}(x) = \frac{x - \widehat{x}_{\frac{1}{2L}}(x)}{\lambda}$ and also that $\|\widehat{x}_{\frac{1}{2L}}(x) - x\| \leq \frac{G}{2L}$. Since $\Phi_{\frac{1}{2L}}(\cdot, x)$ is a $L$-strongly convex and $G$ Lipschitz function in the domain $\left\{ x' : \|\widehat{x}_{\frac{1}{2L}}(x) - x\| \leq \frac{G}{2L} \right\}$, (Harvey et al., 2019, Theorem 3.1) tells us that we can use SGD with $O\left(\frac{G^2 \log \frac{1}{\delta}}{L\widetilde{\epsilon}}\right)$ stochastic gradient oracle queries to implement Step 1 of Algorithm 3 with success probability at least $1 - \delta$, where $\widetilde{\epsilon} := \frac{\epsilon^2}{4L}$. Simplifying this expression gives us a stochastic gradient oracle complexity of $O\left(\frac{G^2 \log \frac{1}{\delta}}{\epsilon^2}\right)$.   □

## D    PROOFS OF RESULTS IN SECTION 4.3

*Proof of Theorem 2.* Letting $h(x, \lambda) := \sum_{i \in [k]} \lambda_i g_i(x)$, we note that $\|\nabla_{xx} h(x, \lambda)\| = \|\sum_{i \in [k]} \lambda_i \nabla^2 g_i(x)\| \leq L(1 + G')^2 + GL'$, where we used Lemma 1 and the fact that $\sum_i |\lambda_i| \leq 1$. On the other hand, $\|\nabla_{x\lambda} h(x, \lambda)\| = \|\sum_{i \in [k]} \nabla g_i(x)\|\| \leq kG(1 + G')$, where we again used Lemma 1. Since $\nabla_{\lambda\lambda} h(x, \lambda) = 0$, we can conclude that $h$ is an $\widehat{L}$-gradient Lipschitz function with $\widehat{L} := L(1 + G')^2 + GL' + kG(1 + G')$. Consequently, $g(x) = \max_{\lambda \in S} h(x, \lambda)$, where $S := \left\{ \lambda \in \mathbb{R}^k : \lambda_i \geq 0, \sum_{i \in [k]} \lambda_i = 1 \right\}$, is $\widehat{L}$-weakly convex and the Moreau envelope $g_{\frac{1}{2\widehat{L}}}$ is well defined.

Denote $\widehat{g}(x, x_s) := \max_{i \in [k]} g_i(x) + \widehat{L}\|x - x_s\|^2$. We now divide the analysis of each of iteration of Algorithm 2 into two cases.

**Case I,** $\widehat{g}(x_{s+1}, x_s) \leq \max_{i \in [k]} g_i(x_s) - 3\widehat{\epsilon}/4$: Since $\max_{i \in [k]} g_i(x_{s+1}) \leq \widehat{g}(x_{s+1}, x_s) \leq \max_{i \in [k]} g_i(x_s) - 3\widehat{\epsilon}/4$, we see that in this case $g(x_s)$ decreases monotonically by at least $3\widehat{\epsilon}/4$ in each iteration. Since by Assumption 1, $g$ is bounded by $B$ in magnitude, and the termination condition in Step 4 guarantees monotonic decrease in every iteration, there can only be at most $2B/(3\widehat{\epsilon}/4) = 8B/\widehat{\epsilon}$ such iterations in Case I.

**Case II,** $\widehat{g}(x_{s+1}, x_s) \geq \max_{i \in [k]} g_i(x_s) - 3\widehat{\epsilon}/4$: In this case, we claim that $x_s$ is an $\epsilon$-FOSP of $g = \max_{i \in [k]} g_i(x)$. To see this, we first note that

$$g(x_s) - 3\widehat{\epsilon}/4 \leq \widehat{g}(x_{s+1}, x_s) \leq (\min_x g(x) + \widehat{L}\|x - x_s\|^2) + \widehat{\epsilon}/4 \implies g(x_s) < \min_x \widehat{g}(x; x_s) + \widehat{\epsilon}. \tag{12}$$

Let $x_s^* := \arg\min_x \widehat{g}(x; x_k)$. Since $g$ is $\widehat{L}$-gradient Lipschitz, we note that $\widehat{g}(\cdot; x_s)$ is $\widehat{L}$-strongly convex. We now use this to prove that $x_s$ is close to $x_s^*$:

$$\widehat{g}(x_s^*; x_s) + \frac{\widehat{L}}{2}\|x_s - x_s^*\|^2 \leq \widehat{g}(x_s; x_s) = f(x_s) \overset{(a)}{<} \widehat{g}(x_k^*; x_s) + \widehat{\epsilon} \implies \|x_s - x_s^*\| < \sqrt{\frac{2\widehat{\epsilon}}{\widehat{L}}} \tag{13}$$

where $(a)$ uses (12). Now consider any $\widehat{x}$, such that $4\sqrt{\widehat{\epsilon}/L} \leq \|\widehat{x} - x_s\|$. Then,

$$g(\widehat{x}) + L\|\widehat{x} - x_s\|^2 = \max_{i \in [k]} g_i(\widehat{x}) + L\|\widehat{x} - x_s\|^2 = \widehat{g}(\widehat{x}; x_s) \overset{(a)}{=} \widehat{g}(x_s^*; x_s) + \frac{L}{2}\|\widehat{x} - x_s^*\|^2$$

$$\overset{(b)}{\geq} f(x_s) - \widehat{\epsilon} + \frac{\widehat{L}}{2}(\|\widehat{x} - x_s\| - \|x_s - x_s^*\|)^2 \overset{(c)}{\geq} f(x_s) + \widehat{\epsilon}, \tag{14}$$

where $(a)$ uses uses $\widehat{L}$-strong convexity of $\widehat{g}(\cdot; x_s)$ at its minimizer $x_s^*$, $(b)$ uses (12), and $(b)$ and $(c)$ use triangle inequality, (13) and $4\sqrt{\widehat{\epsilon}/\widehat{L}} \leq \|\widehat{x} - x_s\|$.

Now consider the Moreau envelope, $g_{\frac{1}{2\widehat{L}}}(x) = \min_{x'} \phi_{\frac{1}{2\widehat{L}}, x}(x')$ where $\phi_{\frac{1}{2\widehat{L}}, x}(x') = g(x') + L\|x - x'\|^2$. Then, we can see that $\phi_{\frac{1}{2\widehat{L}}, x_s}(x')$ achieves its minimum in the ball $\{x' \mid \|x' - x_s\| \leq 4\sqrt{\widehat{\epsilon}/\widehat{L}}\}$ by (14) and Lemma 3(a). Then, with Lemma 3(b,c) and $\widehat{\epsilon} = \frac{\varepsilon^2}{64\widehat{L}}$, we get that,

$$\|\nabla g_{\frac{1}{2\widehat{L}}}(x_s)\| \leq (2\widehat{L})\|x_s - \widehat{x}_{\frac{1}{2\widehat{L}}}(x_s)\| = 8\sqrt{\widehat{L}\widehat{\epsilon}} = \varepsilon, \tag{15}$$

i.e., $x_s$ is an $\varepsilon$-FOSP of $g$.

By combining the above two cases, we establish that $\frac{8B}{3\widehat{\epsilon}}$ "outer" iterations ensure convergence to a $\varepsilon$-FOSP.

We now compute the gradient call complexity of each of these "outer" iterations, where we have two options for implementing Step 3 of Algorithm 2. Note that this step corresponds to solving $\min_x \max_{\lambda \in S} h(x, \lambda)$ up to an accuracy of $\widehat{\epsilon}/4$.

**Option I**, (Thekumparampil et al., 2019, Algorithm 2): Since the minimax optimization problem here is $\widehat{L}$-strongly-convex–concave and $2\widehat{L}$-gradient Lipschitz, (Thekumparampil et al., 2019, Theorem 1) tells us that this requires at most $m$ gradient calls for each $g_i$ where,

$$\frac{6(2\widehat{L})^2}{Lm^2} \leq \frac{\widehat{\epsilon}}{4} = \frac{\varepsilon^2}{2^8 \widehat{L}} \implies O\left(\frac{\widehat{L}}{\varepsilon}\right) \leq m \tag{16}$$

Therefore the number of gradient computations required for each iteration of inner problem is $O\left(\frac{\widehat{L}}{\epsilon} \log^2\left(\frac{1}{\varepsilon}\right)\right)$.

**Option II, Cutting plane method** (Lee et al., 2015): Let us consider $u(\lambda) := \min_x h(x, \lambda) + \widehat{L}\|x - x_s\|^2$, which is a $\widehat{L}$-Lipschitz, concave function of $\lambda$. (Lee et al., 2015) tells us that we can use cutting plane algorithms to obtain $\widehat{\lambda}$ satisfying $u(\widehat{\lambda}) \geq \max_{\lambda \in S} u(\lambda) - \widetilde{\epsilon}$ using $O\left(k \log \frac{k\widehat{L}}{\widetilde{\epsilon}}\right)$ gradient queries to $u$ and $\text{poly}(k \log \frac{\widehat{L}}{\widetilde{\epsilon}})$ computation. The gradient of $u$ is given by $\nabla u(\lambda) = \nabla_\lambda h(x^*(\lambda), \lambda)$, where $x^*(\lambda) := \text{argmin}_x h(x, \lambda) + \widehat{L}\|x - x_s\|^2$. Since $h(x, \lambda) + \widehat{L}\|x - x_s\|^2$ is a $3\widehat{L}$-smooth and $\widehat{L}$-strongly convex function in $x$, $x^*(\lambda)$ can be computed up to $\widetilde{\epsilon}$ error using gradient descent in $O\left(\log \frac{\widehat{L}}{\widetilde{\epsilon}}\right)$ iterations. If we choose $\widetilde{\epsilon} = \widehat{\epsilon}^2/\text{poly}(k, \widehat{L}/\mu)$, then Proposition 2 tells us that $x^*(\widehat{\lambda})$ satisfies the requirements of Step (3) of Algorithm 2 and the total number of gradient calls to each $g_i$ is at most $O\left(\text{poly}(k) \log \frac{\widehat{L}}{\epsilon}\right)$ in each outer iteration of Algorithm 2.

$\square$

**Proposition 2.** *Suppose $h : \mathbb{R}^{d_1} \times \mathcal{U} \to \mathbb{R}$ be such that $h(\cdot, \lambda)$ is $\mu$-strongly convex for every $\lambda \in \mathcal{U}$, $h(x, \cdot)$ is concave for every $x \in \mathbb{R}^{d_1}$ and $h$ is $L$-gradient Lipschitz. Let $\widehat{\lambda}$ be such that $\min_x h(x, \widehat{\lambda}) \geq \max_\lambda \min_x h(x, \lambda) - \epsilon$ and let $x^*(\widehat{\lambda}) := \text{argmin}_x h(x, \widehat{\lambda})$. Then, we have $\max_\lambda h(x^*(\widehat{\lambda}), \lambda) \leq \min_x \max_\lambda h(x, \lambda) + c\left(\frac{L}{\mu} \cdot \epsilon + \frac{LD_\mathcal{U}}{\sqrt{\mu}} \cdot \sqrt{\epsilon}\right)$, where $D_\mathcal{U} = \max_{\lambda_1, \lambda_2 \in \mathcal{U}} \|\lambda_1 - \lambda_2\|$ is the diameter of $\mathcal{U}$.*

*Proof of Proposition 2.* From the hypothesis, we have:

$$\epsilon \geq h(x^*, \lambda^*) - h(x^*(\widehat{\lambda}), \widehat{\lambda}) \geq h(x^*, \widehat{\lambda}) - h(x^*(\widehat{\lambda}), \widehat{\lambda}) \geq \frac{\mu}{2}\|x^* - x^*(\widehat{\lambda})\|^2,$$

where $(x^*, \lambda^*)$ is the Nash equilibrium and the second step follows from the fact that $\lambda^* = \text{argmax}_\lambda h(x^*, \lambda)$ and the third step follows from the fact that $x^*(\widehat{\lambda}) = \text{argmin}_x h(x, \widehat{\lambda})$. Consequently, we have that $\|x^* - x^*(\widehat{\lambda})\| \leq \sqrt{2\epsilon/\mu}$. Let $\bar{\lambda} := \text{argmax}_\lambda h(x^*(\widehat{\lambda}), \lambda)$. We now have

that:

$$\max_\lambda h(x^*(\widehat{\lambda}), \lambda) - \max_\lambda \min_x h(x, \lambda) = h(x^*(\widehat{\lambda}), \bar{\lambda}) - h(x^*, \lambda^*)$$

$$= h(x^*(\widehat{\lambda}), \bar{\lambda}) - h(x^*(\widehat{\lambda}), \lambda^*) + h(x^*(\widehat{\lambda}), \lambda^*) - h(x^*, \lambda^*)$$

$$\overset{(\zeta_1)}{\leq} \langle \nabla_\lambda h(x^*(\widehat{\lambda}), \lambda^*), \bar{\lambda} - \lambda^* \rangle + \frac{L}{2} \|x^*(\widehat{\lambda}) - x^*\|^2$$

$$\overset{(\zeta_2)}{\leq} \|\nabla_\lambda h(x^*(\widehat{\lambda}), \lambda^*)\| \|\bar{\lambda} - \lambda^*\| + \frac{L\epsilon}{\mu}$$

$$\overset{(\zeta_3)}{\leq} \left( \|\nabla_\lambda h(x^*, \lambda^*)\| + L\|x^*(\widehat{\lambda}) - x^*\| \right) \mathcal{D}_\mathcal{U} + \frac{L\epsilon}{\mu}$$

$$\leq \frac{L\mathcal{D}_\mathcal{U}\sqrt{2\epsilon}}{\sqrt{\mu}} + \frac{L\epsilon}{\mu},$$

where $(\zeta_1)$ follows from the fact that $h(x^*(\widehat{\lambda}), \cdot)$ is concave and $x^* = \operatorname{argmin}_x h(x, \lambda^*)$, $(\zeta_2)$ follows from the bound $\|x^* - x^*(\widehat{\lambda})\| \leq \sqrt{2\epsilon/\mu}$, $(\zeta_3)$ follows from the $L$-gradient Lipschitz property of $h$, and the last step follows from the fact that $\nabla_\lambda h(x^*, \lambda^*) = 0$. This proves the proposition. $\qquad \square$

# E    PROOFS OF RESULTS IN SECTION 5

In this appendix, we present the proofs for the lemmas in Section 5. Recall the SGA update:

$$y_{t+1} = y_t + \eta \nabla_y f_{\sigma(t)}(x, y_t). \tag{17}$$

Therefore, the Jacobian of the $T$-step SGA update is given by

$$Dy_{t+1} = \left( I + \eta \nabla_{yy} f_{\sigma(t)}(x, y_t) \right) Dy_t + \eta \nabla_{yx} f_{\sigma(t)}(x, y_t), \text{ with } \mathcal{A}(x, z) = y_T(x). \tag{18}$$

## E.1    PROOF OF THEOREM 3

**Theorem 3** (General Case). *Suppose for all $j \in [n]$, $f_j$ satisfies Assumption 1. Then, for any fixed randomness $z$, $T$-step SGA is $(1 + \eta L)^T$-Lipschitz and $4(\rho/L) \cdot (1 + \eta L)^{2T}$-gradient Lipschitz.*

*Proof.* **Lipschitz of $y_t(x)$.** We first show the Lipschitz claim. We have the following bound on the Jacobian of the update equation given in (18):

$$\|Dy_{t+1}(x)\| \leq \|(I + \eta \nabla_{yy}^2 f(x, y_t(x))) Dy_t(x)\| + \eta \|\nabla_{yx}^2 f(x, y_t(x))\|$$
$$\leq (1 + \eta L) \|Dy_t(x)\| + \eta L.$$

Since $Dy_0(x) = 0$, the above recursion implies that

$$\|Dy_t(x)\| \leq \eta L \sum_{\tau=0}^{t-1} (1 + \eta L)^\tau \leq (1 + \eta L)^t.$$

**Gradient-Lipschitz of $y_t(x)$.** Next, we show the claimed gradient Lipschitz constant. As above, using the update equation in (18), we have the following bound on the Jacobian:

$$\|Dy_{t+1}(x_1) - Dy_{t+1}(x_2)\|$$
$$\leq \|(I + \eta \nabla_{yy}^2 f(x_1, y_t(x_1)))(Dy_t(x_1) - Dy_t(x_2))\| + \eta \|\nabla_{yx}^2 f(x_1, y_t(x_1)) - \nabla_{yx}^2 f(x_2, y_t(x_2))\|$$
$$\quad + \eta \|[\nabla_{yy}^2 f(x_1, y_t(x_1)) - \nabla_{yy}^2 f(x_2, y_t(x_2))] Dy_t(x_2)\|$$
$$\leq (1 + \eta L) \|Dy_t(x_1) - Dy_t(x_2)\| + \eta \rho (1 + \|Dy_t(x_2)\|)(\|x_1 - x_2\| + \|y_t(x_1) - y_t(x_2)\|)$$
$$\leq (1 + \eta L) \|Dy_t(x_1) - Dy_t(x_2)\| + 4\eta \rho (1 + \eta L)^{2t} \|x_1 - x_2\|.$$

The above recursion implies the claimed Lipschitz constant. Indeed,

$$\|Dy_t(x_1) - Dy_t(x_2)\| \leq 4\eta \rho \sum_{\tau=0}^{t-1} (1 + \eta L)^{t+\tau-1} \|x_1 - x_2\| \leq 4(\rho/L) \cdot (1 + \eta L)^{2t} \|x_1 - x_2\|.$$

$$\square$$

### E.2    PROOF OF THEOREM 4

**Theorem 4** (Concave Case). *Suppose for all $j \in [n]$, $f_j$ satisfies Assumption 1 and $f_j(x, \cdot)$ is concave for any $x$. Then, for any fixed randomness $z$, $T$-step SGA is $\eta LT$-Lipschitz and $(\rho/L) \cdot (1 + \eta LT)^3$-gradient Lipschitz.*

*Proof.* **Lipschitz of $y_t(x)$.** We first show the Lipschitz claim. Using the update equation in (18), we have the following bound on the Jacobian:

$$\|Dy_{t+1}(x)\| \le \|(I + \eta \nabla_{yy}^2 f(x, y_t(x))) Dy_t(x)\| + \eta \|\nabla_{yx}^2 f(x, y_t(x))\|$$
$$\le \|Dy_t(x)\| + \eta L.$$

Since $Dy_0(x) = 0$, the above recursive implies that $\|Dy_t(x)\| \le \eta L t$.

**Gradient-Lipschitz of $y_t(x)$.** Next, we show the claimed gradient Lipschitz constant. Using the update equation in (18):, we have the following bound on the Jacobian:

$$\|Dy_{t+1}(x_1) - Dy_{t+1}(x_2)\|$$
$$\le \|(I + \eta \nabla_{yy}^2 f(x_1, y_t(x_1)))(Dy_t(x_1) - Dy_t(x_2))\| + \eta \|\nabla_{yx}^2 f(x_1, y_t(x_1)) - \nabla_{yx}^2 f(x_2, y_t(x_2))\|$$
$$+ \eta \|[\nabla_{yy}^2 f(x_1, y_t(x_1)) - \nabla_{yy}^2 f(x_2, y_t(x_2))] Dy_t(x_2)\|$$
$$\le \|Dy_t(x_1) - Dy_t(x_2)\| + \eta \rho (1 + \|Dy_t(x_2)\|)(\|x_1 - x_2\| + \|y_t(x_1) - y_t(x_2)\|)$$
$$\le \|Dy_t(x_1) - Dy_t(x_2)\| + \eta \rho (1 + \eta L t)^2 \|x_1 - x_2\|.$$

This recursion implies the following gradient Lipschitz constant:

$$\|Dy_t(x_1) - Dy_t(x_2)\| \le \eta \rho \sum_{\tau=0}^{t-1} (1 + \eta L \tau)^2 \|x_1 - x_2\| \le (\rho/L) \cdot (1 + \eta L t)^3 \|x_1 - x_2\|.$$

$\square$

### E.3    PROOF OF THEOREM 5

**Theorem 5** (Strongly-concave Case). *Suppose for all $j \in [n]$, $f_j$ satisfies Assumption 1 and $f_j(x, \cdot)$ is $\alpha$-strongly concave for any $x$. Then, for any fixed randomness $z$, $T$-step SGA is $\kappa$-Lipschitz and $4(\rho/L) \cdot \kappa^3$-gradient Lipschitz, where $\kappa = L/\alpha$ is the condition number.*

*Proof.* Denote the condition number $\kappa = L/\alpha$.

**Lipschitz of $y_t(x)$.** We first show the claimed Lipschitz constant. Using the update equation in (18), we have that

$$\|Dy_{t+1}(x)\| \le \|(I + \eta \nabla_{yy}^2 f(x, y_t(x))) Dy_t(x)\| + \eta \|\nabla_{yx}^2 f(x, y_t(x))\|$$
$$\le (1 - \eta \alpha) \|Dy_t(x)\| + \eta L.$$

Since $Dy_0(x) = 0$, the above recursion gives the following bound:

$$\|Dy_t(x)\| \le \eta L \sum_{\tau=0}^{t-1} (1 - \eta \alpha)^\tau \le \kappa.$$

**Gradient-Lipschitz of $y_t(x)$.** Next we show the claimed gradient Lipschitz constant. Again, using the update equation, we have that

$$\|Dy_{t+1}(x_1) - Dy_{t+1}(x_2)\|$$
$$\le \|(I + \eta \nabla_{yy}^2 f(x_1, y_t(x_1)))(Dy_t(x_1) - Dy_t(x_2))\| + \eta \|\nabla_{yx}^2 f(x_1, y_t(x_1)) - \nabla_{yx}^2 f(x_2, y_t(x_2))\|$$
$$+ \eta \|[\nabla_{yy}^2 f(x_1, y_t(x_1)) - \nabla_{yy}^2 f(x_2, y_t(x_2))] Dy_t(x_2)\|$$
$$\le (1 - \eta \alpha) \|Dy_t(x_1) - Dy_t(x_2)\| + \eta \rho (1 + \|Dy_t(x_2)\|)(\|x_1 - x_2\| + \|y_t(x_1) - y_t(x_2)\|)$$
$$\le (1 - \eta \alpha) \|Dy_t(x_1) - Dy_t(x_2)\| + 4 \eta \rho \kappa^2 \|x_1 - x_2\|.$$

This recursion implies that

$$\|Dy_t(x_1) - Dy_t(x_2)\| \leq 4\eta\rho\kappa^2 \sum_{\tau=0}^{t-1}(1-\eta\alpha)^\tau\|x_1 - x_2\| \leq 4(\rho/L) \cdot \kappa^3\|x_1 - x_2\|.$$

$\square$

### E.4 PROOF OF THEOREM 6

**Theorem 6** (General Case). *Suppose for all $j \in [n]$, $f_j$ satisfies Assumption 1. Then, for any fixed seed $z$, $T$-step SNAG is $T(1 + \eta L/\theta)^T$-Lipschitz and $50(\rho/L) \cdot T^3(1 + \eta L/\theta)^{2T}$-gradient Lipschitz.*

*Proof.* Recall the SNAG update

$$\tilde{y}_t = y_t + (1 - \theta)(y_t - y_{t-1}) \tag{19}$$

$$y_{t+1} = \tilde{y}_t + \eta\nabla_y f_{\sigma(t)}(x, \tilde{y}_t). \tag{20}$$

Observe that the update equation for $T$-step SNAG implies that

$$D\tilde{y}_t = Dy_t + (1 - \theta)(Dy_t - Dy_{t-1})$$
$$Dy_{t+1} = (I + \eta\nabla_{yy}f_{\sigma(t)}(x, \tilde{y}_t))D\tilde{y}_t + \eta\nabla_{yx}f_{\sigma(t)}(x, \tilde{y}_t) \tag{21}$$

**Lipschitz of $y_t(x), v_t(x)$.** We first show the claimed Lipschitz constant. By the update equations in (21), we have that

$$Dy_{t+1} = (I + \eta\nabla_{yy}f_{\sigma(t)}(x, \tilde{y}_t))(Dy_t + (1 - \theta)(Dy_t - Dy_{t-1})) + \eta\nabla_{yx}f_{\sigma(t)}(x, \tilde{y}_t).$$

Denote $\delta_t = \|Dy_t - Dy_{t-1}\|$, and note that $Dy_0 = Dy_{-1} = 0$ so that $\delta_0 = 0$. By the equation above, we have that

$$\delta_{t+1} \leq \eta L\|Dy_t\| + (1 + \eta L)(1 - \theta)\delta_t + \eta L$$
$$\leq \eta L \sum_{\tau=1}^{t}\delta_\tau + (1 + \eta L)(1 - \theta)\delta_t + \eta L.$$

In the following, we use induction to prove that

$$\delta_t \leq (1 + \eta L/\theta)^t. \tag{22}$$

It is easy to verify that this is true for the base case $\delta_0 = 0 \leq 1$. Suppose the claim is true for all $\tau \leq t$, then we have that

$$\delta_{t+1} \leq \eta L \sum_{\tau=1}^{t}(1 + \eta L/\theta)^\tau + (1 + \eta L)(1 - \theta)(1 + \eta L/\theta)^t + \eta L$$
$$\leq \eta L \sum_{\tau=0}^{t}(1 + \eta L/\theta)^\tau + (1 - \theta)(1 + \eta L/\theta)^{t+1}$$
$$= \theta[(1 + \eta L/\theta)^{t+1} - 1] + (1 - \theta)(1 + \eta L/\theta)^{t+1} \leq (1 + \eta L/\theta)^{t+1}.$$

This proves the induction claim. Therefore, by (22), we have the following two bounds:

$$\|Dy_t(x)\| \leq \sum_{\tau=1}^{t}\delta_\tau \leq t(1 + \eta L/\theta)^t,$$

$$\|D\tilde{y}_t(x)\| \leq (2 - \theta)\|Dy_t(x)\| + (1 - \theta)\|Dy_{t-1}(x)\| \leq 3t(1 + \eta L/\theta)^t.$$

**Gradient-Lipschitz of $y_t(x)$.** Next, we show the claimed gradient Lipschitz constant. For any fixed $x_1, x_2$, denote $w_t = Dy_t(x_1) - Dy_t(x_2)$, we have

$$w_{t+1} = (I + \eta\nabla_{yy}f_{\sigma(t)}(x_1, \tilde{y}_t(x_1)))(w_t + (1 - \theta)(w_t - w_{t-1}))$$
$$+ \underbrace{\eta(\nabla_{yx}f_{\sigma(t)}(x_1, \tilde{y}_t(x_1)) - \nabla_{yx}f_{\sigma(t)}(x_2, \tilde{y}_t(x_2)))}_{T_1}$$
$$+ \underbrace{\eta(\nabla_{yy}f_{\sigma(t)}(x_1, \tilde{y}_t(x_1)) - \nabla_{yy}f_{\sigma(t)}(x_2, \tilde{y}_t(x_2)))(Dy_t(x_2) + (1 - \theta)(Dy_t(x_2) - Dy_{t-1}(x_2)))}_{T_2}.$$

We note that we can upper bound the last two terms above as follows:

$$
\begin{aligned}
T_1 + T_2 \leq & \eta\rho(\|x_1 - x_2\| + \|\tilde{y}_t(x_1) - \tilde{y}_t(x_2)\|) \\
& + \eta\rho(\|x_1 - x_2\| + \|\tilde{y}_t(x_1) - \tilde{y}_t(x_2)\|)(2\|Dy_t(x_2)\| + \|Dy_{t-1}(x_2)\|) \\
\leq & 24\eta\rho t^2 (1 + \eta L/\theta)^{2t} \|x_1 - x_2\|.
\end{aligned}
$$

Therefore, let $\zeta_t = \|w_t - w_{t-1}\|$, and $\Delta = \|x_1 - x_2\|$, we have the following:

$$
\begin{aligned}
\zeta_{t+1} \leq & \eta L \|w_t\| + (1 + \eta L)(1 - \theta)\zeta_t + 24\eta\rho t^2 (1 + \eta L/\theta)^{2t}\Delta \\
\leq & \eta L \sum_{\tau=1}^{t} \zeta_\tau + (1 + \eta L)(1 - \theta)\zeta_t + 24\eta\rho t^2 (1 + \eta L/\theta)^{2t}\Delta.
\end{aligned}
$$

In the following, we use induction to prove that

$$
\zeta_t \leq 50(\rho/L) \cdot t^2 (1 + \eta L/\theta)^{2t}\Delta := \psi(t). \tag{23}
$$

It is easy to verify that this is true for the base case $\zeta_0 = 0$. Suppose the claim is true for all $\tau \leq t$, then we have that

$$
\begin{aligned}
\zeta_{t+1} \leq & 50\eta\rho \sum_{\tau=1}^{t} \tau^2 (1 + \eta L/\theta)^{2\tau}\Delta + (1 + \eta L)(1 - \theta)\psi(t) + 24\eta\rho t^2(1 + \eta L/\theta)^{2t}\Delta \\
\leq & \theta(\rho/L) \cdot [50t^2 \frac{(1 + \eta L/\theta)^{2(t+1)} - 1}{(1 + \eta L/\theta)^2 - 1} + 24(\eta L/\theta)t^2(1 + \eta L/\theta)^{2t}]\Delta + (1 - \theta)\psi(t+1) \\
\leq & \theta(\rho/L) \cdot [25t^2(1 + \eta L/\theta)^{2(t+1)} + 24t^2(1 + \eta L/\theta)^{2(t+1)}]\Delta + (1 - \theta)\psi(t+1) \\
\leq & \theta(\rho/L) \cdot [50t^2(1 + \eta L/\theta)^{2(t+1)}]\Delta + (1 - \theta)\psi(t+1) \leq \psi(t+1).
\end{aligned}
$$

This proves the induction claim. Therefore, by (23), we have that

$$
\|Dy_t(x_1) - Dy_t(x_2)\| = \|w_t\| \leq \sum_{\tau=1}^{t} \zeta_\tau \leq 50(\rho/L)t^3(1 + \eta L/\theta)^{2t}\|x_1 - x_2\|.
$$

$\square$

### E.5 PROJECTED GRADIENT ASCENT IS NOT GRADIENT-LIPSCHITZ

**Proposition 3.** *Consider $f(x, y) = xy$ for $(x, y) \in \mathcal{X} \times \mathcal{Y}$, where $\mathcal{X} = [0, 10]$ and $\mathcal{Y} = [0, 1]$.*
*1-step projected gradient ascent given by:*

$$
y_1(x) = \mathcal{P}_\mathcal{Y}(y_0 + \eta \nabla_y f(x, y_0)) \quad y_0 = 0,
$$

*where $\eta > 1/10$ is not a gradient-Lipschitz algorithm.*

*Proof.* We see that $y_1(x) = \min(1, \eta x)$ and $f(x, y_1(x)) = x \min(1, \eta x)$. For $\eta < 1/10$, we see that $f(x, y_1(x))$ is not gradient-Lipschitz at $x = 1/\eta \in (0, 10)$. $\square$

## F ON NONCONVERGENCE OF EXTRAGRADIENT METHOD AND OPTIMISTIC GRADIENT DESCENT ASCENT (OGDA)

In this section, we show that there exist nonconvex-nonconcave problems where the extragradient method and OGDA get trapped in limit cycles. While the example we construct is a quadratic function that is not bounded over all of the space, it will be clear from the proof that the same results hold for any function that matches the constructed quadratic locally around origin.

Consider the following two dimensional quadratic function parametrized by $a, b \in \mathbb{R}$:

$$
f(x, y) = \frac{1}{2}ax^2 + bxy - \frac{1}{2}ay^2 = \frac{1}{2}\begin{pmatrix} x & y \end{pmatrix}\begin{pmatrix} a & b \\ b & -a \end{pmatrix}\begin{pmatrix} x \\ y \end{pmatrix} \tag{24}
$$

Denote $z := (x, y)$, $\Phi := \begin{pmatrix} a & b \\ -b & a \end{pmatrix}$ and define:

$$F(z) := \begin{pmatrix} \nabla_x f(x, y) \\ -\nabla_y f(x, y) \end{pmatrix} = \underbrace{\begin{pmatrix} a & b \\ -b & a \end{pmatrix}}_{\Phi} \begin{pmatrix} x \\ y \end{pmatrix} := \Phi \cdot \begin{pmatrix} x \\ y \end{pmatrix},$$

Let the eigenvalues of matrix $\Phi$ be $\lambda_1, \lambda_2$. We know they satify the characteristic equation $(\lambda - a)^2 + b^2 = 0$. By Vieta's formulas, we know that $\lambda_1 + \lambda_2 = 2a$, $\lambda_1 \lambda_2 = a^2 + b^2$. This justifies the following proposition.

**Proposition 4.** *For any complex number $c \in \mathbb{C}$, there exist $a, b \in \mathbb{R}$ s.t. the eigenvalues of $\Phi$ are $c$ and $\bar{c}$. Here $\bar{c}$ is the complex conjugate of $c$.*

*Proof.* Let $c = p + qi$ for some $p, q \in \mathbb{R}$, we can choose $a = (c + \bar{c})/2 = p$, $b = \pm\sqrt{c\bar{c} - a^2} = \pm q$. It is easy to check such matrix $\Phi$ has eigenvalues $c$ and $\bar{c}$. $\square$

### F.1    EXTRAGRADIENT METHODS

The extragradient algorithm (Korpelevich, 1976) uses the following update:

$$z_{t+1/2} = z_t - \eta F(z_t)$$
$$z_{t+1} = z_t - \eta F(z_{t+1/2})$$

For the case of quadratic function specified by (24), this update is equivalent to

$$z_{t+1} = (I - \eta\Phi(I - \eta\Phi))z_t = (I - \eta\Phi + \eta^2\Phi^2)z_t$$

Clearly, the eigenvalues of matrix $(I - \eta\Phi + \eta^2\Phi^2)$ are simply $(1 - \eta\lambda_1 + \eta^2\lambda_1^2)$ and $(1 - \eta\lambda_2 + \eta^2\lambda_2^2)$.

**Proposition 5.** *For the extragradient algorithm with any fixed learning rate $\eta$, there exists a quadratic function, and appropriate initial points such that the extragradient algorithm is trapped in a limit cycle.*

*Proof.* For any given $\theta \in (0, 2\pi)$, we can simply pick $\lambda_1 \in \mathbf{C}$ to be the solution of equation $1 - \eta\lambda_1 + \eta^2\lambda_1^2 = e^{i\theta}$. This is a quadratic equation of $\lambda_1$, which always has nonzero complex roots. For $\lambda_2 = \bar{\lambda_1}$, we verify that $1 - \eta\lambda_2 + \eta^2\lambda_2^2 = e^{-i\theta}$. Finally, by Proposition 4, we know that there exists a choice of $a, b$ such that, the two eigenvalues of $(I - \eta\Phi + \eta^2\Phi^2)$ are precisely $e^{i\theta}$ and $e^{-i\theta}$. If the initial point $\begin{pmatrix} x_0 \\ y_0 \end{pmatrix}$ can be decomposed as $\begin{pmatrix} x_0 \\ y_0 \end{pmatrix} = \alpha_1 z_1 + \alpha_2 z_2$, where $z_1$ and $z_2$ are the two eigenvectors of $\Phi$ corresponding to eigenvalues $\lambda_1$ and $\lambda_2$ respectively, we see that the extra gradient algorithm follows the trajectory $\begin{pmatrix} x_t \\ y_t \end{pmatrix} = \alpha_1 e^{i\theta t} z_1 + \alpha_2 e^{-i\theta t} z_2$, thereby getting trapped in a limit cycle. $\square$

### F.2    OPTIMISTIC GRADIENT DESCENT ASCENT (OGDA)

The OGDA algorithm (Daskalakis & Panageas, 2018a) has the following update:

$$z_{t+1} = z_t - 2\eta F(z_t) + \eta F(z_{t-1}).$$

(Daskalakis & Panageas, 2018a)[Section 4.1] presents an example where OGDA is empirically not shown to converge for some initialization points. Here, we present an example for which nonconvergence of OGDA is shown rigorously. We can rewrite this update as

$$\begin{pmatrix} z_{t+1} \\ z_t \end{pmatrix} = \begin{pmatrix} I - 2\eta\Phi & \eta\Phi \\ I & 0 \end{pmatrix} \begin{pmatrix} z_t \\ z_{t-1} \end{pmatrix}$$

There are four eigenvalues for the matrix $\begin{pmatrix} I - 2\eta\Phi & \eta\Phi \\ I & 0 \end{pmatrix}$. Two of them $\mu_1, \mu_2$ are eigenvalues of the matrix $\begin{pmatrix} 1 - 2\eta\lambda_1 & \eta\lambda_1 \\ 1 & 0 \end{pmatrix}$ while the other two $\mu_3, \mu_4$ are eigenvalues of the matrix $\begin{pmatrix} 1 - 2\eta\lambda_2 & \eta\lambda_2 \\ 1 & 0 \end{pmatrix}$, where $\lambda_1$ and $\lambda_2$ are the eigenvalues of $\Phi$.

**Proposition 6.** *For the OGDA algorithm with any fixed learning rate $\eta$, there exists a quadratic function, and a choice of initial points such that the iterates are trapped in a limit cycle.*

*Proof.* We will fist show that there is a choice for $a$ and $b$ so that the resulting eigenvalues $\mu_1$, $\mu_2$ of $\begin{pmatrix} 1 - 2\eta\lambda_1 & \eta\lambda_1 \\ 1 & 0 \end{pmatrix}$ and $\mu_3$, $\mu_4$ of $\begin{pmatrix} 1 - 2\eta\lambda_2 & \eta\lambda_2 \\ 1 & 0 \end{pmatrix}$ satisfy $|\mu_1| = |\mu_3| = 1$ and $|\mu_2| = |\mu_4| < 1$. Let us first fix $\mu_1 = e^{i\theta}$ and choose $\eta\lambda_1 = \frac{\mu_1(\mu_1-1)}{2\mu_1-1}$ so that we have $\mu_1$ is a root of $\mu^2 - (1 - 2\eta\lambda_1)\mu + \eta\lambda_1 = 0$. We note that $\mu_2$ is the other root of the above equation. We have that:

$$|\eta\lambda_1| = \frac{|\mu_1 - 1|}{|2\mu_1 - 1|} = \sqrt{\frac{(\cos\theta - 1)^2 + \sin^2\theta}{(2\cos\theta - 1)^2 + \sin^2\theta}}.$$

Therefore, for any $\theta \in (0, \pi/4]$, we have $|\eta\lambda_1| < 1$. That is $|\mu_1\mu_2| < 1$, since $|\mu_1| = 1$, we have $|\mu_2| < 1$. By symmetry, we know by choosing $\lambda_2 = \bar{\lambda}_1$, the two eigenvalues of $\begin{pmatrix} 1 - 2\eta\lambda_2 & \eta\lambda_2 \\ 1 & 0 \end{pmatrix}$ are $\mu_3 = \bar{\mu}_1$ and $\mu_4 = \bar{\mu}_2$. In sum, we have proved that for any $\theta \in (0, \pi/4]$, and any learning rate $\eta$, there exists a choice of $a, b$ such that the four eigenvalues of $\begin{pmatrix} I - 2\eta\Phi & \eta\Phi \\ I & 0 \end{pmatrix}$ are $e^{\pm i\theta}, c', \bar{c}'$ where $|c'| = |\bar{c}'| < 1$. If the initial points $\begin{pmatrix} z_1 \\ z_0 \end{pmatrix}$ have the decomposition $\sum_{i=1}^{4} \alpha_i \bar{z}_i$, where $\bar{z}_i$ is the eigenvector corresponding to eigenvalue $\mu_i$, then the iterates $z_t$ and $z_{t-1}$ are given by $\begin{pmatrix} z_t \\ z_{t-1} \end{pmatrix} = \sum_{i=1}^{4} \alpha_i \mu_i^t \bar{z}_i$. As $t \to \infty$, $\begin{pmatrix} z_t \\ z_{t-1} \end{pmatrix}$ converges to $\alpha_1 e^{i\theta t} \bar{z}_1 + \alpha_3 e^{-i\theta t} \bar{z}_3$, proving that OGDA iterates get trapped in a limit cycle and neither converge to a single point nor escape to infinity. $\qquad\square$

## G  ADDITIONAL EXPERIMENTS AND DETAILS

In this appendix section, we provide additional experimental results and details.

**Dirac-GAN.** In the results presented in Section 6 for this problem, the discriminator sampled its initialization uniformly from the interval $[-0.1, 0.1]$ and performed $T = 10$ steps of gradient ascent. For the results given in Figure 5, we allow the discriminator to sample uniformly from the interval $[-0.5, 1]$ and consider the discriminator performing $T = 100$ (Figure 5b) and $T = 1000$ (Figure 5c) gradient ascent steps. The rest of the experimental setup is equivalent to that described in Section 6.

For the result presented in Figure 5b, we see that with this distribution of initializations for the discriminator and $T = 100$ gradient ascent steps, the generator is not able to converge to the optimal parameter of $\theta^* = 0$ to recreate the underlying data distribution using our algorithm which descends $\nabla f(\theta, \mathcal{A}(\theta))$ or the algorithm that descends $\nabla_\theta f(\theta, \mathcal{A}(\theta))$. However, we see that our algorithm converges significantly closer to the optimal parameter configuration. Furthermore, we still observe stability and convergence from our training method, whereas standard training methods using simultaneous or alternating gradient descent-ascent always cycle. This example highlights that the optimization through the algorithm of the adversary is important not only for the rate of convergence, but it also influences what the training method converges to and gives improved results in this regard.

Finally, in the result presented in Figure 5b, we see that with this distribution of initializations for the discriminator and $T = 1000$ gradient ascent steps, the generator is able to converge to the optimal parameter of $\theta^* = 0$ to recreate the underlying data distribution using our algorithm which descends $\nabla f(\theta, \mathcal{A}(\theta))$ or the algorithm that descends $\nabla_\theta f(\theta, \mathcal{A}(\theta))$. Thus, while with $T = 100$ we did not observe convergence to the optimal generator parameter, with a stronger adversary we do see convergence to the optimal generator parameter. This behavior can be explained by the fact that when the discriminator is able to perform enough gradient ascent steps to nearly converge, the gradients $\nabla f(\theta, \mathcal{A}(\theta))$ and $\nabla_\theta f(\theta, \mathcal{A}(\theta))$ are nearly equivalent.

We remark that we repeated the experiments 5 times with different random seeds and show the mean generator parameters during the training with a window around the mean of a standard deviation. The results were very similar between runs so the window around the mean is not visible.

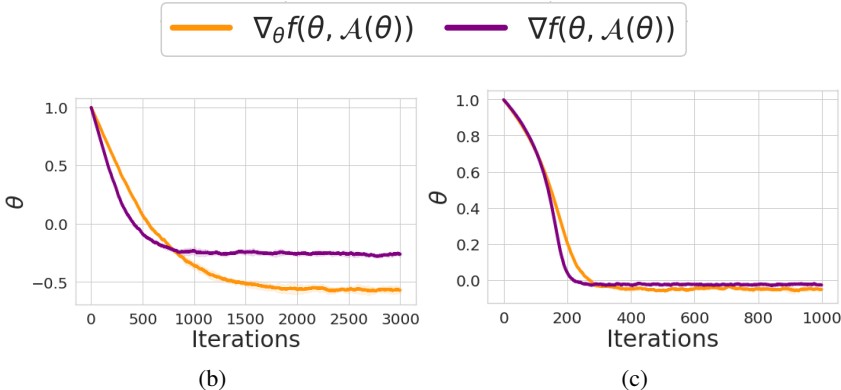

Figure 5: **Dirac-GAN**: Generator parameters while training using our framework with and without optimizing through the discriminator where between each generator update the discriminator samples an initial parameter choice uniformly at random from the interval $[-0.5, 1]$ and then performs $T = 100$ (Figure 5b) and $T = 1000$ (Figure 5c) steps of gradient ascent.

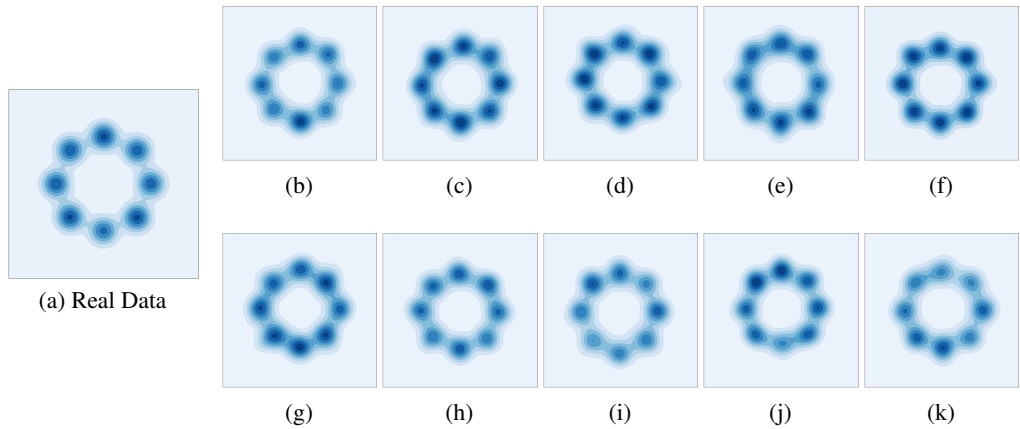

Figure 6: **Mixture of Gaussians**: Figure 6a shows the real data distribution and Figures 6b–6k show the final generated distributions after 150k generator updates from 10 separate runs of the training procedure described in Section 6 using gradient ascent for the discriminator. Each run recovers a generator distribution closely resembling the underlying data distribution.

**Mixture of Gaussians.** We noted in Section 6 that we repeated our experiment training a generative adversarial network to learn a mixture of Gaussians 10 times and observed that for each run of the experiment our training algorithm recovered all modes of the distribution. We now show those results in Figure 6. In particular, in Figure 6a we show the real data distribution and in Figures 6b–6k we show the final generated distribution from 10 separate runs of the training procedure after 150k generator updates. Notably, we observe that each run of the training algorithm is able to generate a distribution that closely resembles the underlying data distribution, showing the stability and robustness of our training method.

We also performed an experiment on the mixture of Gaussian problem in which the discriminator algorithm was the Adam optimization procedure with parameters $(\beta_1, \beta_2) = (0.99, 0.999)$ and learning rate $\eta_2 = 0.004$ and the generator learning rate was $\eta_1 = 0.05$. The rest of the experimental setup remained the same. We ran this experiment 10 times and observed that for 7 out of the 10 runs of the final generated distribution was reasonably close to the real data distribution, while for 3 out of the 10 runs the generator did not learn the proper distribution. This is to say that we found the training algorithm was not as stable when the discriminator used Adam versus normal gradient ascent. The final generated distribution from the 7 of 10 runs with reasonable distributions are shown in Figure 7.

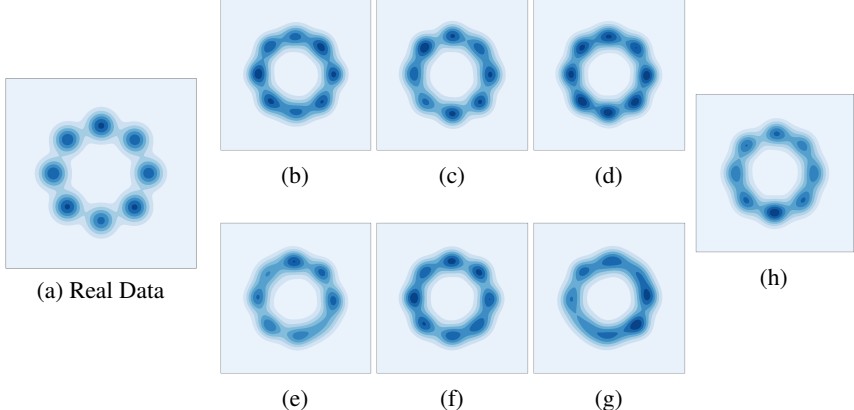

Figure 7: **Mixture of Gaussians**: Figure 7a shows the real data distribution and Figures 7b–7h show the final generated distributions after 150k generator updates from the 7 out of 10 separate runs of the training procedure using Adam optimization for the discriminator that produced reasonable distributions.

**Adversarial Training.** We now provide some further background on the adversarial training experiment and additional results. It is now well-documented that the effectiveness of deep learning classification models can be vulnerable to adversarial attacks that perturb the input data (see, e.g., Biggio et al. 2013; Szegedy et al. 2014; Kurakin et al. 2017; Madry et al. 2018). A common approach toward remedying this vulnerability is by training the classification model against adversarial perturbations. Recall from Section 6 that given a data distribution $\mathcal{D}$ over pairs of examples $x \in \mathbb{R}^d$ and labels $y \in [k]$, parameters $\theta$ of a neural network, a set $\mathcal{S} \subset \mathbb{R}^d$ of allowable adversarial perturbations, and a loss function $\ell(\cdot, \cdot, \cdot)$ dependent on the network parameters and the data, adversarial training amounts to considering a minmax optimization problem of the form $\min_\theta \mathbb{E}_{(x,y)\sim\mathcal{D}}[\max_{\delta \in \mathcal{S}} \ell(\theta, x + \delta, y)]$.

A typical approach to solving this problem is an alternating optimization approach (Madry et al., 2018). In particular, each time a batch of data is drawn from the distribution, $T$-steps of projected gradient ascent are performed by ascending along the sign of the gradient of the loss function with respect to the data and projecting back onto the set of allowable perturbations, then the parameters of the neural network are updated by descending along the gradient of the loss function with the perturbed examples. The experimental setup we consider is analogous but the inner maximization loop performs $T$-steps of regular gradient ascent (not using the sign of the gradient and without projections).

For the adversarial training experiment considered in Section 6, we also evaluated the trained models against various other attacks. Recall that the models were training using $T = 10$ steps of gradient ascent in the inner optimization loop with a learning rate of $\eta_2 = 4$. To begin, we evaluated the trained models against gradient ascent attacks with a fixed learning rate of $\eta_2 = 4$ and a number of steps $T \in \{5, 10, 20, 40\}$. We also evaluated the trained models against gradient ascent attacks with a fixed budget of $T\eta_2 = 40$ and various choices of $T$ and $\eta_2$. These results are presented in Figure 9. Finally, we evaluated the trained models against attacks using the Adam optimization method with a fixed budget of $T\eta_2 = 0.04$ and various choices of $T$ and $\eta_2$. These results are presented in Figure 10. Notably, we see that our algorithm outperforms the baselines and similar conclusions can be drawn as from the experiments for adversarial training presented in Section 6. The additional experiments highlight that our method of adversarial training is robust against attacks that the algorithm did not use in training when of comparable computational power and also that it improves robustness against attacks of greater computational power than used during training.

In Section 6, we showed the results of evaluating the gradient norms $\|\nabla f(\theta, \mathcal{A}(\theta))\|$ as a function of the number of gradient ascent steps $T$ in the adversary algorithm $\mathcal{A}$ and observed that it grows much slower than exponentially. Here we provide more details on the setup. We took a run of our algorithm trained with the setup described in Section 6 and retrieved the models that were saved after 25, 50, 75, and 100 training epochs. For each model, we then sampled 100 minibatches of data and

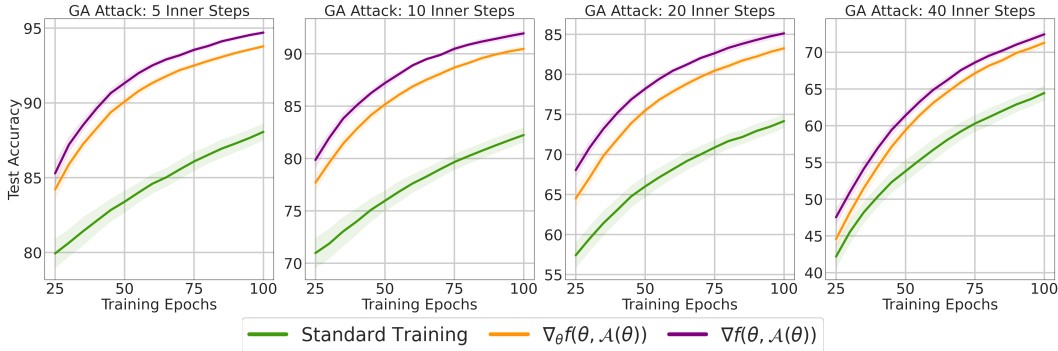

Figure 8: **Adversarial Training**: Test accuracy during the course of training against gradient ascent attacks with a fixed learning rate of $\eta_2 = 4$ and the number of steps $T \in \{5, 10, 20, 40\}$.

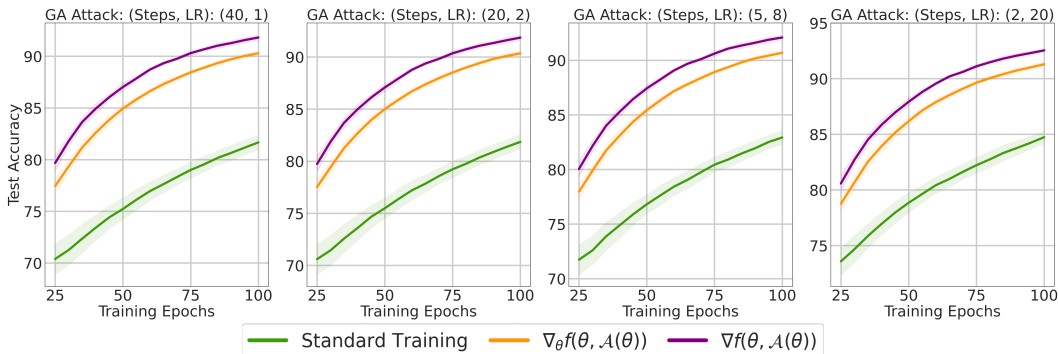

Figure 9: **Adversarial Training**: Test accuracy during the course of training against gradient ascent attacks with a fixed attack budget of $T\eta_2 = 40$ where $T$ is the number of attack steps and $\eta_2$ is the learning rate (LR).

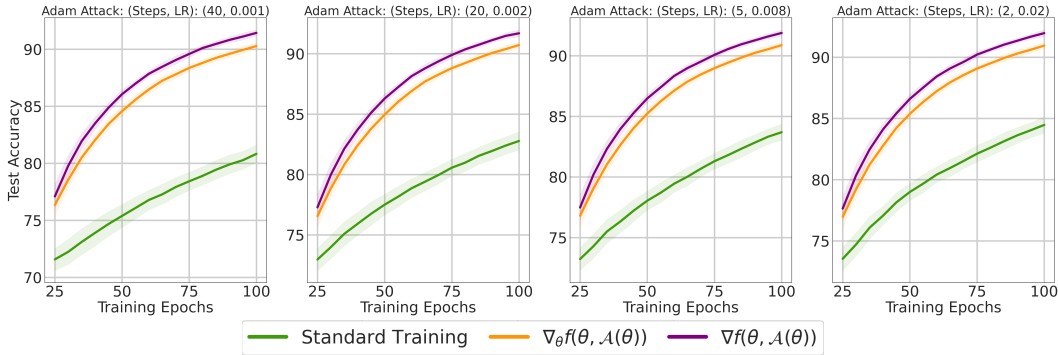

Figure 10: **Adversarial Training**: Test accuracy during the course of training against Adam optimization attacks with a fixed attack budget of $T\eta_2 = 0.04$ where $T$ is the number of attack steps and $\eta_2$ is the learning rate (LR).

for each minibatch performed $T \in \{20, 30, 40, 50, 60, 70, 80, 90, 100\}$ steps of gradient ascent with learning rate $\eta_2 = 4$ (the learning rate from training) and then computed the norm of the gradient $\nabla f(\theta, \mathcal{A}(\theta))$ where $\mathcal{A}$ corresponds to the gradient ascent procedure with the given number of steps and learning rate. In Figure 2, which is reproduced here in Figure 11a, the mean of the norm of the gradients over the sampled minibatches are shown with the shaded window indicating a standard deviation around the mean. We also repeated this procedure using $\eta_2 = 1$ and show the results in Figure 11b from which similar conclusions can be drawn.

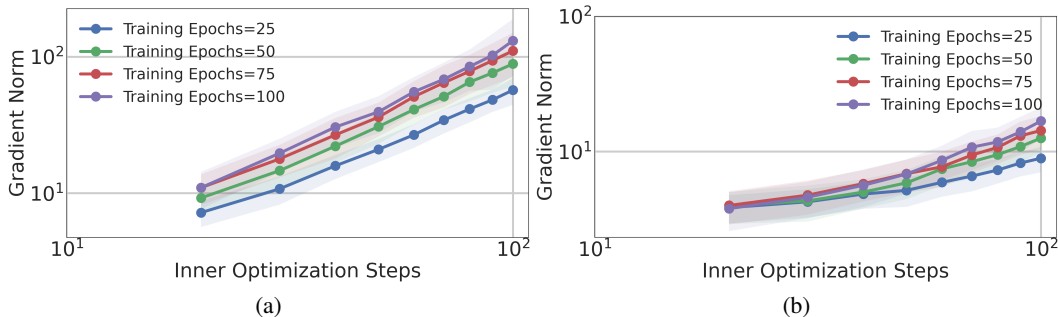

Figure 11: **Adversarial Training**: $\|\nabla f(\theta, \mathcal{A}(\theta))\|$ as a function of the number of steps $T$ taken by the gradient ascent algorithm $\mathcal{A}$ evaluated at multiple points in the training procedure. Figure 11a corresponds to using $\eta_2 = 4$ in the gradient ascent procedure and Figure 11b corresponds to using $\eta_2 = 1$ in the gradient ascent procedure.

| Layer Type | Shape |
|---|---|
| Convolution + ReLU | $5 \times 5 \times 20$ |
| Max Pooling | $2 \times 2$ |
| Convolution + ReLU | $5 \times 5 \times 20$ |
| Max Pooling | $2 \times 2$ |
| Fully Connected + ReLU | 800 |
| Fully Connected + ReLU | 500 |
| Softmax | 10 |

Table 1: Convolutional neural network model for the adversarial training experiments.

Finally, we provide details on the convolutional neural network model for the adversarial training experiments. In particular, this model is exactly the same as considered in (Nouiehed et al., 2019) and we reproduce it in Table 1.

**Experimental Details.** For the experiments with neural network models we used two Nvidia GeForce GTX 1080 Ti GPU and the PyTorch higher library(Deleu et al., 2019) to compute $\nabla f(\theta, \mathcal{A}(\theta))$. In total, running all the experiments in the paper takes about half of a day with this computational setup. The code for the experiments is included in the supplementary material with instructions on how to run. We built our code for the adversarial training based off code openly provided by the authors of (Nouiehed et al., 2019).

