# OpenReview forum: "Minimax Optimization with Smooth Algorithmic Adversaries"
_ICLR.cc/2022/Conference — ICLR 2022 Poster_

### Official Review · Reviewer_WVv5 · 2021-10-31

**Correctness:** 3
**Technical Novelty And Significance:** 3
**Empirical Novelty And Significance:** 3
**Recommendation:** 6
**Confidence:** 5

**Main Review:**

Firstly, I would like to mention some technical details in the proofs of the primary appendix sections.

Issue #1.[Lemma 2] In the lemma about the max of L-weak convex functions and its subgradient, the proof shows only the inclusion not the completeness. More precisely, from the proof I can see why the linear combinations of gradients correspond to subgradients of the max-functions. On the other hand, this does not show that all subgradients can be written in this form?

Please provide a new proof OR correct the statement and explain why a refined statement suffices for our task. A priori, I believe that the result should be correct as a generalization of Danskin's Theorem.

Issue #2.[Lemma 3] The statement of the lemma is correct since they are folklore truths for proximal descent. HOWEVER, the proof that you provide has some mathematical flaws.
Example: You say that min g(x) = min g_\ell(x) and then you say that  " Thus argmin g(x) = argmin g_\ell(x)". In general this is not correct for two function f_1,f_2 (i.e f_1(x)=|x|,f_2=|x-1|)
The only way that I know to show the result is via the facts that $argmin g(x) = fixpoints(proxGD)$ and
$argmin g_\ell(x) = fixpoints(proxGD)$

Similarly for item (c), I was not able to parse the argument. Could you please explain in a list the statements of the proof

MAIN CONTRIBUTION:

I really like the idea of the "smoothed adversary" as a refined benchmark. i agree that a better framework would be if we "constrain" the opponent in order to examine if it is possible to find efficiently a good solution. I appreciate the model and the conceptual, "easily"-understandable results and the well-written part of main paper. I don't like the fact that authors do not explain(hide) explicitly that the smooth adversaries have not memory in comparison with its opponent. If this is correct, I would appreciate some addition in the text which will explain that. Finally, there are two issues about the results:
(1) we don't have convergence to a Nash equilibrium or local \eps-Nash equilibrium but to a point which is biased against the "inner" player. (2) The novelty of the results is limited since the algorithmic concepts ,that are used, are somehow blackboxes. On the other hand, I understand that the main idea is to explain why the case of "smooth adversaries" is a tractable case.

In general I would like some answers in my "complaints"/ "issues" that I raised. I believe strongly that with a satisfying answer I will increase happily my score.

**Summary Of The Paper:**

The paper is trying to address the problem of non-convex non-concave min-max optimization under the perspective of application of smoothed algorithms between the two opponents. More precisely,
the paper examine a model where the max-player applied a zero-memory smooth (from differential perspective) algorithm and min-player SGD/SNAG or proximal methods providing results similar with the state of art. The convergence is reassured by "potential" arguments but the fixed point is not an equally game-theoretic notion like Nash Equilibrium.

**Summary Of The Review:**

While there are some typos and restriction, I strongly appreciate the model design which consists the backbone of the paper. Shortly describing, I like the main message of the paper even if there are main details that should be clarified and corrected to express the exact result that can be derived from the paper.

---

> ### Author Response · Authors · 2021-11-12
> **Thank you for the review**
>
> - **Issue 1**:
> Proposition A.22 from Bertsekas’ Thesis:
> Let $f:\mathbb{R}^n\times \mathbb{R}^m\to (-\infty,\infty]$ be a function and let $Y$ be a compact subset of $\mathbb{R}^m$. Assume further that for every vector $y\in Y$ the function $f(\cdot,y):\mathbb{R}^n\to (-\infty,\infty]$ is a closed proper convex function. Consider the function $g$ defined as $g(x)=\sup_{y\in Y}f(x,y)$. Then if $f$ is finite somewhere it is a closed proper convex function. Furthermore if $\text{int}(\text{dom} g)\neq \emptyset$ and $g$ is continuous on the set  $\text{int}(\text{dom} g)\times Y$, then for every $x\in \text{int}(\text{dom} f)$ we have $\partial g(x)=\text{conv}(\partial f(x,\bar{y})|\ \bar{y}\in \overline{Y}(x))$ where $\overline{Y}(x)$ is the set $\overline{Y}(x)=\text{Set}(\bar{y}\in Y|\ g(x,\bar{y})=\max_{y\in Y}g(x,y))$.
>
> The above proposition (which is an extension of Danskin’s theorem by Bertsekas) implies the lemma in our paper. Namely, we claim the following:
>
> Let $g_1,\ldots, g_k:X\to(-\infty,+\infty]$ be convex functions. For every $x\in \cap_{i=1}^k\text{dom}(g_i)$ such that each $g_i$ is continuous at $x$, we have that
> $\partial \max(g_1,\ldots, g_k)(x)=\text{conv}(\cup_{i=1}^k\partial g_i(x))$.
>
> To see that this claim holds let $g(\cdot):=\max_{i\in[k]} g_i(\cdot)$. Define $h:\mathbb{R}^n\times X\to (-\infty,+\infty]$ to be the function $h(\lambda,x):=\sum_{i=1}^k\lambda_ig_i(x)$. Then, for each $x\in X$, we have that $g(x)=\sup_{\lambda\in \Delta_k}h(\lambda,x)$. Moreover, $\partial_x h(\lambda,x)=\sum_{i=1}^k\lambda_i\partial g_i(x)$. From here the Proposition A.22 above implies the result.
>
> We will update the proof accordingly. Thanks for catching this.
>
> - **Issue 2**:
> Yes, thanks for catching this. We have now added a full proof of $\arg\min_xg(x)=\arg\min_xg_\lambda(x)$ in the proof of Lemma 3 in the revised version, which is indeed specific for the Moreau envelope.
>
> - **Item (c) in Lemma 3**: We have updated the proof in the revised version to make this clearer. We are not including the proof here since we had some trouble in getting Latex markup in the comment working properly.
>
> - **Lack of memory**: While we did mention at multiple places, especially in the empirical results section, that the discriminator is initialized from scratch for every generator update, we notice now that it is not mentioned prominently in the introduction. We will describe this point in the introduction in the revised version.
> - **Convergence to Nash equilibria**: Since Nash equilibria are not guaranteed to exist in general nonconvex-nonconcave problems [1], we cannot in general expect convergence to such points. Nevertheless, both prior results on adversarial models [2] as well as our empirical results, suggest that having bias against smooth adversaries is still a practically useful property and motivate our theoretical investigation and results.
> - **Algorithmic concepts are blackboxes**: We do not fully understand the concern here. If it is that we allow the adversary to use arbitrary algorithms for inner maximization, we note that since our algorithm differentiates through the adversary’s algorithm, it still adapts to the nature of adversary’s algorithm. Please let us know if our understanding of your concern is incorrect.
>
> [1] What is Local Optimality in Nonconvex-Nonconcave Minimax Optimization? by Jin et al. ICML 2020
>
> [2] Do Adversarially Robust ImageNet Models Transfer Better? by Salman et al. NeurIPS 2020

---

### Official Review · Reviewer_nxeu · 2021-11-01

**Correctness:** 3
**Technical Novelty And Significance:** 3
**Empirical Novelty And Significance:** 3
**Recommendation:** 6
**Confidence:** 4

**Main Review:**

Some comments:

1) The authors should provide the gap between the original nonconvex-nonconcave minimax problem $\min_x max_y f(x,y)$ and the obtained nonconvex-concave minimax problem $min_x max_{i\in [k]}f(x,\mathcal{A}_i(x))$ in both the theoretical and experimental aspects.

2) Assumption 1 given in the paper may be not mild. So the authors should detail whether these assumptions are used in the existing methods.

3) Recently many existing methods have studied the nonconvex-concave minimax problems. The authors should point the differences between the nonconvex-concave minimax problem (1) given in the paper and the generic nonconvex-concave minimax problems studied in the existing methods.

4) The authors study the convergence properties of the proposed algorithms to solve the nonconvex-concave minimax problem (1) given in the paper. Compared the existing studies on the generic nonconvex-concave minimax problems, what are challenges?


**Summary Of The Paper:**

This paper provides a new perspective to solve the nonconvex-nonconcave minimax problems. Specifically, the proposed method turns the nonconvex-nonconcave minimax problem to a nonconvex-concave minimax problem by using a toolkit of multiple smooth algorithms to find a solution of the maximization problem given fixed $x$. Moreover, it studies the convergence properties of the proposed algorithms. Some experimental results on generative gdversarial networks and adversarial training demonstrate the efficiency of the proposed algorithms.

**Summary Of The Review:**

Overall, I like this paper. I hope the authors can deal with the above comments.

---

> ### Author Response · Authors · 2021-11-12
> **Thanks for the review**
>
> - **gap between the original nonconvex-nonconcave minimax problem and the obtained nonconvex-concave minimax problem**: Given that the nonconvex-nonconcave minimax problem is NP-hard, while our nonconvex-concave reformulation is tractable, in the worst case, the gap between the original version and reformulation could be large. However, our empirical results on GANs and adversarial training suggest that reformulation still leads to desired solutions.
> - **Assumption 1**: Boundedness (a), Lipschitzness (b) and gradient Lipschitzness (c) assumptions are standard in nonconvex optimization for finding first order stationary points. In our context, the additional Hessian Lipschitzness (d) assumption is required only for ensuring popular algorithms such as SGA etc. to also be smooth (Theorems 3-5).
> - **General nonconvex-concave vs our formulation/algorithm**: The key specialization in our formulation, compared to recent works on general nonconvex-concave optimization, is that in our case, the maximization is linear and over the simplex. In terms of algorithms and results: Theorem 1 uses existing ideas and results while both Algorithm 2 and the $O(\epsilon^{-2})$ convergence rate in Theorem 2 are novel and improve over existing results on nonconvex-concave minimax optimization.

---

### Official Review · Reviewer_BUCz · 2021-11-02

**Correctness:** 4
**Technical Novelty And Significance:** 3
**Empirical Novelty And Significance:** 3
**Recommendation:** 8
**Confidence:** 4

**Main Review:**

1) Their algorithms need $O(\epsilon^4)$ gradient queries. It would be great if the authors can clarify that is this tight bound or can we achieve better complexity and if we can what would be a proper approach.

2)The setting is novel and interesting from a practical and theoretical point of view.

3)Author motivates the subject from a theoretical and practical point of view.

4)In the introduction, the author reviewed the previous works carefully.

5)The authors successfully provide simulation results for their algorithm.

6)The paper is well-organized and well-written.

7)the code is included.


**Summary Of The Paper:**

This paper focuses on solving minimax optimization when the maximization oracle has a limited amount of computation power and can not compute the max exactly. They provide SGD based and proximal algorithms and analyze them for weakly convex function.

**Summary Of The Review:**

1) Their algorithms need $O(\epsilon^4)$ gradient queries. It would be great if the authors can clarify that is this tight bound or can we achieve better complexity and if we can what would be a proper approach.

2)The setting is novel and interesting from a practical and theoretical point of view.

3)Author motivates the subject from a theoretical and practical point of view.

4)In the introduction, the author reviewed the previous works carefully.

5)The authors successfully provide simulation results for their algorithm.

6)The paper is well-organized and well-written.

7)the code is included.

---

> ### Author Response · Authors · 2021-11-12
> **Thank you for the review**
>
> We do not know if the convergence rate of $O(\epsilon^{-4})$ is optimal. It is an interesting question to figure out the optimal rate of convergence for finding FOSP of problems of the form $\min_x \max_i g_i(x)$.

---

### Official Review · Reviewer_9PEU · 2021-11-04

**Correctness:** 4
**Technical Novelty And Significance:** 3
**Empirical Novelty And Significance:** 3
**Recommendation:** 8
**Confidence:** 3

**Main Review:**

The topic of the paper is quite relevant in the community. The main idea makes sense and the experimental results are quite impressive.


Technical Issue:
- I am afraid that the gradient of an algorithm is not well-defined. How do you compute the gradient? Probably the paper assumes that the algorithm trains a neural network and its gradient is that of the network. Please clarify it.

**Summary Of The Paper:**

The paper considers a min-max optimization of general nonconvex functions. The paper proposes methods to approximately find stationary points by "smoothing" the max part. Experimental results show the superiority of the proposed approach to previous work.

**Summary Of The Review:**

The paper is good enough and it is supported by experimental results.

---

> ### Author Response · Authors · 2021-11-12
> **Thank you for the review**
>
> As described just below Definition 1, we define the gradient of a deterministic algorithm by looking at it as a function that takes inputs to outputs. For a randomized algorithm, we consider it as a distribution over deterministic algorithms and work with gradients of these deterministic algorithms.

---

### Author Response · Authors · 2021-11-18
**Gentle reminder about our response**

Dear reviewers, a gentle reminder to please look at our responses and let us know if you have any further questions/concerns.

---

### Decision · Program_Chairs · 2022-01-20

**Decision:**

Accept (Poster)

**Comment:**

The paper addresses the problem of non-convex non-concave min-max optimization under the perspective of application of smoothed algorithms between two opponents.
The paper examines a model where the max-player applied a zero-memory smooth (from differential perspective) algorithm and min-player SGD/SNAG or proximal methods providing results similar with the state-of-art. Convergence guarantees proposed were sound and experimental results on generative adversarial networks and adversarial training demonstrate the efficiency of the proposed algorithms.